# Enhancing Adversarial Robustness via Score-Based Optimization

Boya Zhang [*]        Weijian Luo [†]        Zhihua Zhang [‡]

## Abstract

Adversarial attacks have the potential to mislead deep neural network classifiers by introducing slight perturbations. Developing algorithms that can mitigate the effects of these attacks is crucial for ensuring the safe use of artificial intelligence. Recent studies have suggested that score-based diffusion models are effective in adversarial defenses. However, existing diffusion-based defenses rely on the sequential simulation of the reversed stochastic differential equations of diffusion models, which are computationally inefficient and yield suboptimal results. In this paper, we introduce a novel adversarial defense scheme named *ScoreOpt*, which optimizes adversarial samples at test-time, towards original clean data in the direction guided by score-based priors. We conduct comprehensive experiments on multiple datasets, including CIFAR10, CIFAR100 and ImageNet. Our experimental results demonstrate that our approach outperforms existing adversarial defenses in terms of both robustness performance and inference speed.

## 1   Introduction

In recent years, there have been breakthrough performance improvements with deep neural networks (DNNs), particularly in the realms of image classification, object detection, and semantic segmentation, as evidenced by the works of [31, 59, 53, 19, 13, 51]. However, DNNs have been shown to be easily deceived to produce incorrect predictions by simply adding human-imperceptible perturbations to inputs. These perturbations are called adversarial attacks [14, 36, 38, 7], resulting in safety concerns. Therefore, studies of mitigating the impact of adversarial attacks on DNNs, which is referred to as *adversarial defenses*, are significant for AI safety.

Various strategies have been proposed in order to enhance the robustness of DNN classifiers against adversarial attacks. One of the most effective forms of adversarial defense is *adversarial training* [36, 71, 16, 32, 71, 44], which involves training a classifier with both clean data and adversarial samples. However, adversarial training has its limitations as it necessitates prior knowledge of the specific attack method employed to generate adversarial examples, thus rendering it inadequate in handling previously unseen types of adversarial attacks or corruptions.

On the contrary, *adversarial purification* [48, 68, 52, 28, 69, 40, 21] is another form of promising defense that leverages a standalone purification model to eliminate adversarial signals before conducting downstream classification tasks. The primary benefit of adversarial purification is that it obviates the need to retrain the classifier, enabling adaptive defense against adversarial attacks at *test time*. Additionally, it showcases significant generalization ability in purifying a wide range of adversarial attacks, without affecting pre-existing natural classifiers. The integration of adversarial purification models into AI systems necessitates minor adjustments, making it a viable approach to enhancing the robustness of DNN-based classifiers.

---

[*]Academy for Advanced Interdisciplinary Studies; Peking University; `zhangboya@pku.edu.cn`;

[†]School of Mathematical Sciences; Peking University; `luoweijian@stu.pku.edu.cn`;

[‡]School of Mathematical Sciences; Peking University; `zhzhang@math.pku.edu.cn`;

37th Conference on Neural Information Processing Systems (NeurIPS 2023).

Figure 1: Illustration of our proposed adversarial defense framework.

Diffusion models [22, 54, 58], also known as score-based generative models, have demonstrated state-of-the-art performance in various applications, including image and audio generation [9, 30], molecule design [23], and text-to-image generation [39]. Apart from their impressive generation ability, diffusion models have also exhibited the potential to improve the robustness of neural networks against adversarial attacks. Specifically, they can function as adaptive test-time purification models [40, 16, 4, 63, 66].

Diffusion-based purification methods have shown great success in improving adversarial robustness, but still have their own limitations. For instance, they require careful selection of the appropriate hyper-parameters such as forward diffusion timestep [40] and guidance scale [63], which can be challenging to tune in practice. In addition, diffusion-based purification relies on the simulation of the underlying stochastic differential equation (SDE) solver. The reverse purification process requires iteratively denoising samples step by step, leading to heavy computational costs [40, 63].

In the hope of circumventing the aforementioned issues, we introduce a novel adversarial defense scheme that we call *ScoreOpt*. Our key intuition is to derive the posterior distribution of clean samples given a specific adversarial example. Then adversarial samples can be optimized towards the points that maximize the posterior distribution with gradient-based algorithms at test time. The prior knowledge is provided by pre-trained diffusion models. Our defense is independent of base classifiers and applicable across different types of adversarial attacks, making it flexible enough across various domains of applications. The illustration of our method is presented in Figure 1.

Our main contributions can be summarized in three aspects as follows:

- We propose a novel adversarial defense scheme that optimizes adversarial samples to reach the points with the local maximum likelihood of the posterior distribution that is defined by pre-trained score-based priors.
- We explore effective loss functions for the optimization process, introduce a novel score regularizer and propose corresponding practical algorithms.
- We conduct extensive experiments to demonstrate that our method not only achieves start-of-the-art performance on various benchmarks but also improves the inference speed.

## 2 Preliminary

### 2.1 Score-based Diffusion Models

Score-based diffusion models [22, 58] learn how to transform complex data distributions to relatively simple ones such as the Gaussian distribution, and vice versa. Diffusion models consist of two processes, a forward process that adds Gaussian noise to input $\mathbf{x}$ from $\mathbf{x}_0$ to $\mathbf{x}_T$, and a reverse generative process that gradually removes random noise from a sample until it is fully denoised. For the continuous-time diffusion models (see Song et al. [58] for more details and further discussions), we can use two SDEs to describe the above data-transformation processes, respectively. The forward-time SDE is given by:

$$d\mathbf{x}_t = \mathbf{f}(\mathbf{x}_t, t)dt + \mathbf{g}(t)d\mathbf{w}_t, \ t \in [0, T];$$

where $\mathbf{f} : \mathbb{R}^D \to \mathbb{R}^D$ and $\mathbf{g} : \mathbb{R} \to \mathbb{R}$ are drift and diffusion coefficients respectively, $\mathbf{w}$ denotes the standard Wiener process. While new samples are generated by solving the reverse-time SDE:

$$d\mathbf{x}_t = [\mathbf{f}(\mathbf{x}_t, t) - \mathbf{g}^2(t)\nabla_{\mathbf{x}_t} \log p_t(\mathbf{x}_t)]dt + \mathbf{g}(t)d\bar{\mathbf{w}}_t, t \in [T, 0];$$

where $\bar{\mathbf{w}}$ defines the reverse-time Wiener process. The score function term $\nabla_{\mathbf{x}_t} \log p_t(\mathbf{x}_t)$ is usually parameterized by a time-dependent neural network $s_{\boldsymbol{\theta}}(\mathbf{x}_t; t)$ and trained by score-matching related techniques [25, 58, 61, 56, 41].

**Diffusion Models as Prior** Recent studies have investigated the incorporation of an additional constraint to condition diffusion models for high-dimensional data sampling. DDPM-PnP [17] proposed transforming diffusion models into plug-and-play priors, allowing for parameterized samples, and utilizing diffusion models as critics to optimize over image space. Building on the formulation in Graikos et al. [17], DreamFusion [43] further introduced a more stable training process. The main idea is to leverage a pre-trained 2D image diffusion model as a prior for optimizing a parameterized 3D representation model. The resulting approach is called score distillation sampling (SDS), which bypasses the computationally expensive backpropagation through the diffusion model itself by simply excluding the score-network Jacobian term from the diffusion model training loss. SDS uses the approximate gradient to train a parametric NeRF generator efficiently. Other works [35, 37, 62] followed similar approaches to extend SDS to the latent space of latent diffusion models [46].

## 2.2 Diffusion-based Adversarial Purification

Diffusion models have also gained significant attention in the field of adversarial purification recently. They have been employed not only for empirical defenses against adversarial attacks [69, 40, 63, 66], but also for enhancing certified robustness [5, 67]. The unified procedure for applying diffusion models in adversarial purification involves two processes. The forward process adds random Gaussian noise to the adversarial example within a small diffusion timestep $t^*$, while the reverse process recovers clean images from the diffused samples by solving the reverse-time stochastic differential equation. Implementing the aforementioned forward-and-denoise procedure, imperceptible adversarial signals can be effectively eliminated. Under certain conditions, the purified sample restores the original clean sample with a high probability in theory [40, 67].

However, diffusion-based purification methods suffer from two main drawbacks. Firstly, their robustness performance heavily relies on the choice of the forward diffusion timestep, denoted as $t^*$. Selecting an appropriate $t^*$ is crucial because excessive noise can lead to the removal of semantic information from the original example, while insufficient noise may fail to eliminate the adversarial perturbation effectively. Secondly, the reverse process of these methods involves sequentially applying the denoising operation from timestep $t$ to the previous timestep $t - 1$, which requires multiple deep network evaluations. In contrast to previous diffusion-based purification methods, our optimization framework departs from the sequential step-by-step denoising procedure.

## 3 Methodology

In this section, we present our proposed adversarial defense scheme in detail. Our method is motivated by solving an optimization problem of adversarial samples to remove the applied attacks. Therefore, we start by formally formulating the optimization objective, followed by exploring effective loss functions and introducing two practical algorithms.

### 3.1 Problem Formulation

Our main idea is to formulate the adversarial defense as an optimization problem given the perturbed sample and the pre-trained prior, in which the solution to the optimization problem is the recovered original sample that we want. We regard the adversarial example $\mathbf{x}_a$ as a disturbed measurement of the original clean example $\mathbf{x}$, and we assume that the clean example is generated by a prior probability distribution $\mathbf{x} \sim p(\mathbf{x})$. The posterior distribution of the original sample given the adversarial example is $p(\mathbf{x}|\mathbf{x}_a) \propto p(\mathbf{x}) \, p(\mathbf{x}_a|\mathbf{x})$. The maximum a posteriori estimator that maximizes the above conditional distribution is given by:

$$\hat{\mathbf{x}}^* = \arg\min_{\mathbf{x}} - \log p(\mathbf{x} \mid \mathbf{x}_a). \tag{1}$$

In this work, we use the data distribution under pretrained diffusion models as the prior $p_{\boldsymbol{\theta}}(\mathbf{x})$.

## 3.2 Loss Functions for Optimization Process

Following Graikos et al. [17], we introduce a variational posterior $q(\mathbf{x})$ to approximate the true posterior distribution $p(\mathbf{x}|\mathbf{x}_a)$ in the original optimization objective. The variational upper bound on the negative log-likelihood $-\log p(\mathbf{x}_a)$ is:

$$-\log p(\mathbf{x}_a) \leq \mathbb{E}_{q(\mathbf{x})} \left[ -\log p(\mathbf{x}_a|\mathbf{x}) \right] + \mathrm{KL}\left( q(\mathbf{x}) \| p_{\boldsymbol{\theta}}(\mathbf{x}) \right). \tag{2}$$

As shown in Song et al. [57] and Vahdat et al. [60], we can further obtain an upper bound on the second Kullback-Leibler (KL) divergence term between the target variational posterior distribution $q(\mathbf{x})$ and the prior distribution defined by the pre-trained diffusion models $p_{\boldsymbol{\theta}}(\mathbf{x})$:

$$\mathrm{KL}\left( q(\mathbf{x}) \| p_{\boldsymbol{\theta}}(\mathbf{x}) \right) \leq \mathbb{E}_{q(\mathbf{x})} \mathbb{E}_{t \sim \mathcal{U}(0,1), \epsilon \sim \mathcal{N}(\mathbf{0}, \mathbf{I})} \left[ w(t) \| s_{\boldsymbol{\theta}}(\mathbf{x}_t; t) - \nabla_{\mathbf{x}_t} \log q(\mathbf{x}_t|\mathbf{x}) \|_2^2 \right], \tag{3}$$

where $w(t) = g(t)^2/2$ is a time-dependent weighting coefficient, $\mathbf{x}_t = \mathbf{x} + \sigma_t \epsilon$ denotes the forward diffusion process, $\sigma_t$ is the pre-designed noise schedule, and $s_{\boldsymbol{\theta}}$ represents the pre-trained diffusion models.

The simplest approximation to the posterior is using a point estimate, i.e., the introduced variational posterior $q(\mathbf{x})$ satisfies the Dirac delta distribution $q(\mathbf{x}) = \delta(\mathbf{x} - \mathbf{x}_\mu)$. Thus, the above upper bound can be rewritten as:

$$\mathbb{E}_{t \sim \mathcal{U}(0,1), \epsilon \sim \mathcal{N}(\mathbf{0}, \mathbf{I})} \left[ w(t) \| s_{\boldsymbol{\theta}}(\mathbf{x}_t; t) - \nabla_{\mathbf{x}_t} \log q(\mathbf{x}_t|\mathbf{x}_\mu) \|_2^2 \right], \tag{4}$$

We simply use notation $\mathbf{x}$ instead of $\mathbf{x}_\mu$ throughout for convenience. The weighted denoising score matching objective in (4) is also equivalent to the diffusion model training loss [43].

According to Tweedie's formula: $\mu_z = z + \Sigma_z \nabla_z \log p(z)$, where $\Sigma$ denotes covariance matrix, we can obtain $\mathbf{x} = \mathbf{x}_t + \sigma_t^2 \nabla_{\mathbf{x}_t} \log q(\mathbf{x}_t)$. Defining $D_{\boldsymbol{\theta}}(\mathbf{x}_t; t) := \mathbf{x}_t + \sigma_t^2 s_{\boldsymbol{\theta}}(\mathbf{x}_t; t)$, the KL term of our opimization objective converts to:

$$\mathbb{E}_{t \sim \mathcal{U}(0,1), \epsilon \sim \mathcal{N}(\mathbf{0}, \mathbf{I})} \left[ \tilde{w}(t) \| D_{\boldsymbol{\theta}}(\mathbf{x} + \sigma_t \epsilon; t) - \mathbf{x} \|_2^2 \right], \tag{5}$$

where $\tilde{w}(t) = w(t)/\sigma_t^2$. Note that $D_{\boldsymbol{\theta}}$ can be used to estimate the denoised image directly, which is called *one-shot* denoiser [33, 5].

In our work, we adopt the approach of setting $\tilde{w}(t) = 1$ for convenience and performance as in previous studies [22, 17]. Since we have no information about the conditional distribution $p(\mathbf{x}_a|\mathbf{x})$, we need a heuristic formulation for the first reconstruction term in (2). The simplest method is to initialize $\mathbf{x}$ by the adversarial sample $\mathbf{x}_a$ and use the loss in (5) to optimize over $\mathbf{x}$ directly, eliminating the constraint term. The rationale behind this simplification is that leading $\mathbf{x}_a$ towards the mode of the $p_{\boldsymbol{\theta}}(\mathbf{x})$ with the same ground-truth class label makes it easier for the natural classifier to produce a correct prediction. In this way, the loss function of our optimization process reduces to:

$$\mathcal{L}_{\mathrm{Diff}}(\mathbf{x}, \boldsymbol{\theta}) = \mathbb{E}_{t \sim \mathcal{U}(0,1), \epsilon \sim \mathcal{N}(\mathbf{0}, \mathbf{I})} \left[ \| D_{\boldsymbol{\theta}}(\mathbf{x} + \sigma_t \epsilon; t) - \mathbf{x} \|_2^2 \right]. \tag{6}$$

Randomized smoothing techniques typically assume that the adversarial perturbation follows a Gaussian distribution. Suppose the Gaussian assumption holds, we can instantiate the reconstruction term with the mean squared error (MSE) between $\mathbf{x}$ and $\mathbf{x}_a$. The optimization objective converts to the following MSE loss:

$$\mathcal{L}_{\mathrm{MSE}}(\mathbf{x}, \mathbf{x}_a, \boldsymbol{\theta}) = \mathbb{E}_{t \sim \mathcal{U}(0,1), \epsilon \sim \mathcal{N}(\mathbf{0}, \mathbf{I})} \left[ \| D_{\boldsymbol{\theta}}(\mathbf{x} + \sigma_t \epsilon; t) - \mathbf{x} \|_2^2 + \lambda \| \mathbf{x} - \mathbf{x}_a \|_2^2 \right], \tag{7}$$

where $\lambda$ is a weighting hyper-parameter to balance the two loss terms.

### 3.2.1 Score Regularization Loss

However, both Diff and MSE optimizations have their own drawbacks. Regarding the Diff loss, the optimization process solely focuses on the score prior $p_{\boldsymbol{\theta}}$, and the update direction guided by the pre-trained diffusion models leads the samples towards the modes of the prior, gradually losing the semantic information of the original samples. As illustrated in Figure 2a, with a sufficient number of optimization steps, both standard and robust accuracies decline significantly. On the other hand,

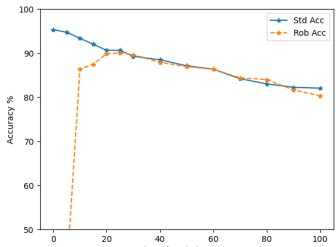 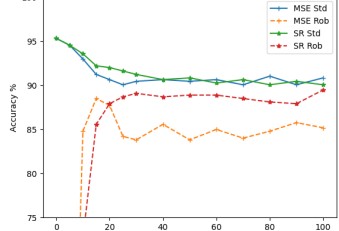 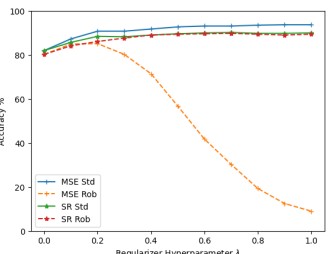

(a) Robustness performance via Diff optimization using different number of optimization steps.

(b) Comparison between MSE and SR with different optimization iterations.

(c) Comparison between MSE and SR with different regularizer hyperparameters.

Figure 2: Robustness performance comparison for Diff, MSE, and SR optimizations.

the MSE loss maintains a high standard accuracy at the cost of a large drop in robust accuracy, as depicted in Figure 2b. Furthermore, Figure 2c demonstrates that the performance of MSE is highly dependent on the weighting hyper-parameter, which controls the intensity of the constraint term.

To address the above-mentioned issues, we propose to introduce a hyperparameter-free score regularization (SR) loss:

$$
\mathcal{L}_{\text{SR}}(\mathbf{x}, \mathbf{x}_a, \boldsymbol{\theta}) = \mathbb{E}_{t \sim \mathcal{U}(0,1), \epsilon_1, \epsilon_2 \sim \mathcal{N}(\mathbf{0}, \mathbf{I})}
$$
$$
\left[ \left\| D_{\boldsymbol{\theta}}\left(\mathbf{x} + \sigma_t \epsilon_1; t\right) - \mathbf{x} \right\|_2^2 + \left\| D_{\boldsymbol{\theta}}\left(\mathbf{x} + \sigma_t \epsilon_1; t\right) - D_{\boldsymbol{\theta}}\left(\mathbf{x}_a + \sigma_t \epsilon_2; t\right) \right\|_2^2 \right]. \quad (8)
$$

We use the introduced constraint to minimize the pixel-level distance between the denoised versions of the current sample $\mathbf{x}$ and initial adversarial sample $\mathbf{x}_a$. The additional regularization term can be expanded as:

$$
\left\| D_{\boldsymbol{\theta}}\left(\mathbf{x} + \sigma_t \epsilon_1; t\right) - D_{\boldsymbol{\theta}}\left(\mathbf{x}_a + \sigma_t \epsilon_2; t\right) \right\|_2^2 \approx \left\| \mathbf{x} - \mathbf{x}_a + \sigma_t^2 \left(s_{\boldsymbol{\theta}}\left(\mathbf{x}_t; t\right) - s_{\boldsymbol{\theta}}\left(\mathbf{x}_{a,t}; t\right)\right) \right\|_2^2. \quad (9)
$$

The SR loss encourages consistency with the original sample in terms of not only the pixel values but also the score function estimations at the given noise level $\sigma_t$. Since the two parts of SR loss correspond to the same noise magnitude $t$ of score networks, there is no need to introduce an additional hyperparameter $\lambda$ as in (7).

Figure 2b and 2c demonstrate the effectiveness of the SR loss. As the number of optimization steps increases, both the standard and robust accuracy converge to stable values, with the latter remaining close to the optimal. In contrast to the MSE loss, the SR loss shows insensitivity to the weighting hyperparameter and significantly outperforms MSE, particularly for larger values of $\lambda$.

### 3.3 Practical Algorithms

---

**Algorithm 1:** (ScoreOpt-O) Optimizing adversarial sample towards robustness with score-based prior.

---

**Input:** Adversarial image $\mathbf{x}_a$, pre-trained score-based diffusion model $s_\theta$, noise level range $[t_{min}, t_{max}]$, optimization iteration steps M, learning rate $\eta$.

$\mathbf{x}_0 = \mathbf{x}_a$;
**for** $i \in 0, ..., M-1$ **do**
    Sample $t \sim \mathcal{U}(t_{min}, t_{max})$, $\epsilon_1, \epsilon_2 \sim \mathcal{N}(\mathbf{0}; \mathbf{I})$;
    $\mathbf{x}_{i,t} = \mathbf{x}_i + \sigma_t \epsilon_1$;
    $\mathbf{x}_{a,t} = \mathbf{x}_a + \sigma_t \epsilon_2$;
    Calculate the gradient grad of Diff loss (6) or SR loss (8) with respect to $\mathbf{x}_i$;
    $\text{grad} = \nabla_{\mathbf{x}_i} \left[ \left\| D_{\boldsymbol{\theta}}\left(\mathbf{x}_{i,t}; t\right) - \mathbf{x}_i \right\|_2^2 + \left\| D_{\boldsymbol{\theta}}\left(\mathbf{x}_{i,t}; t\right) - D_{\boldsymbol{\theta}}\left(\mathbf{x}_{a,t}; t\right) \right\|_2^2 \right]$;
    $\mathbf{x}_{i+1} = \mathbf{x}_i - \eta \cdot \text{grad}$;
**end**
**return** *Purified image* $\mathbf{x}_M$.

---

**Noise Schedule** The perturbation introduced by adversarial attacks is often subtle and challenging for human perception. As a result, our optimization loop does not necessarily incorporate the complete

noise levels employed by the original diffusion models, which generate images from pure random noise. In fact, high noise levels will disrupt local structures and remove semantic information from the input image, leading to a decrease in accuracy. Many previous studies have focused on lower noise levels to conduct image editing tasks. Therefore, we center our pre-designed noise schedule $\sigma_t$ around lower noise levels to preserve the details of the original image as much as possible. Previous diffusion-based purification methods iteratively denoise the noisy image step by step (from $\mathbf{x}_t$ to $\mathbf{x}_{t-1}$), following a predetermined noise schedule. In contrast, our optimization process does not rely on a sequential denoising schedule. Instead, at each iteration, we have the flexibility to randomly select a noise level. This approach allows us to concurrently explore different noise levels during the optimization process.

**Update Rule** Given a noise level $\sigma_t$, we can utilize the aforementioned loss functions to compute the update direction of $\mathbf{x}$. Noting that the objectives correspond to one single random chosen noise magnitude, we propose an alternative update rule to further improve inference speed and robustness performance. We can use the loss gradient to optimize over $\mathbf{x}_t$ directly and then obtain the denoised image $\mathbf{x}$ using one-shot denoising. The loss gradients with respect to $\mathbf{x}$ and $\mathbf{x}_t$ are equivalent in our forward process formulation. Experiments in Section 4 demonstrate that our approach significantly improves one-shot denoising with only a few optimization iterations. These two distinct update rules correspond to Algorithm 1 and Algorithm 2, respectively. Both algorithms are straightforward, effective, and easy to implement.

---

**Algorithm 2:** (ScoreOpt-N) Optimizing noisy adversarial samples and one-shot denoising.

**Input:** Adversarial image $\mathbf{x}_a$, pre-trained score-based diffusion model $s_\theta$, noise level range $[t_{min}, t_{max}]$, optimization iteration steps M and N, learning rate $\eta$.

$\mathbf{x}_0 = \mathbf{x}_a$;
**for** $i \in 0, ..., M-1$ **do**
    *// i denotes the i-th iteration of the outer loop.*
    Sample $t \sim \mathcal{U}(t_{min}, t_{max})$, $\epsilon_1 \sim \mathcal{N}(\mathbf{0}; \mathbf{I})$;
    $\mathbf{x}_{0,t} = \mathbf{x}_i + \sigma_t \epsilon_1$;
    **for** $j \in 0, ..., N-1$ **do**
        *// j denotes the j-th iteration of the inner loop.*
        Sample $\epsilon_2 \sim \mathcal{N}(\mathbf{0}; \mathbf{I})$;
        $\mathbf{x}_{a,t} = \mathbf{x}_a + \sigma_t \epsilon_2$;
        Calculate the gradient grad of Diff loss (6) or SR loss (8) with respect to $\mathbf{x}_{j,t}$;
        $\text{grad} = \nabla_{\mathbf{x}_{j,t}} \left[ \|D_\theta(\mathbf{x}_{j,t}; t) - \mathbf{x}_i\|_2^2 + \|D_\theta(\mathbf{x}_{j,t}; t) - D_\theta(\mathbf{x}_{a,t}; t)\|_2^2 \right]$;
        $\mathbf{x}_{j+1,t} = \mathbf{x}_{j,t} - \eta \cdot \text{grad}$;
    **end**
    *// One-shot denoising.*
    $\mathbf{x}_{i+1} = \mathbf{x}_{N,t} + \sigma_t^2 s_\theta(\mathbf{x}_{N,t}; t)$;
**end**
**return** *Purified image* $\mathbf{x}_M$.

---

## 4 Experiments

This section presents the experimental evaluations conducted to assess the robustness of our proposed methodology against different adversarial attacks, across a range of datasets. Due to space constraints, we defer comprehensive details of the experimental settings, quantitative analysis, and additional results to the Appendix. For all experiments, we report the average results (and corresponding standard deviations) over 5 trials.

### 4.1 Experimental Settings

**Attacks** In this work, we evaluate the robustness performance of adversarial defenses mainly on $\ell_p$-norm bounded threat models. $\epsilon_p$ denotes the perturbation budget under the $\ell_p$-norm. All of the attacks are based on Projected Gradient Descent (PGD) attack [36], a commonly used gradient-based white-box attack, which iteratively perturbs input images along the direction of the gradient of the classifier

Table 1: Standard and robust accuracy against BPDA+EOT attack under $\ell_\infty(\epsilon = 8/255)$ threat model on CIFAR10, compared with other preprocessor-based adversarial defenses and adversarial training methods against white-box attacks.

| Type | Architecture | Method | Standard (%) | Robust (%) | Avg. (%) |
|---|---|---|---|---|---|
| Base Classifier | WRN-28-10 | - | 95.32 | 0.0 | - |
| Adversarial Training | ResNet18 | Madry et al. [36] | 87.30 | 45.80 | 66.55 |
| | | Zhang et al. [71] | 84.90 | 45.80 | 65.35 |
| | WRN-28-10 | Carmon et al. [6] | 89.67 | 63.10 | 76.39 |
| | | Gowal et al. [15] | 89.48 | 64.08 | 77.28 |
| Adversarial Purification | ResNet18 | Yang et al. [68] | 94.80 | 40.80 | 67.80 |
| | ResNet62 | Song et al. [55] | 95.00 | 9.00 | 52.00 |
| | | Hill et al. [20] | 84.12 | 54.90 | 69.51 |
| | | Yoon et al. [69] | 86.14 | 70.01 | 78.08 |
| | WRN-28-10 | Wang et al. [63] | 93.50 | 79.83 | 86.67 |
| | | Nie et al. [40] | 89.02 | 81.40 | 85.21 |
| | | Ours($o$) | 90.78±0.40 | **82.85±0.26** | **86.82** |
| | | Ours($n$) | 93.44±0.40 | **90.59±0.08** | **92.02** |

loss function and projects into some $l_p\epsilon$-ball: $\delta \leftarrow \Pi_\epsilon (\delta + \alpha \cdot \text{sign} (\nabla_x \text{Loss}(f(x + \delta), y)))$, where $f$ represents a classifier.

**Datasets and Baselines**  Three datasets are considered in our experiments: CIFAR10, CIFAR100, and ImageNet. Following previous works, the evaluation is conducted on the test set of each dataset. We adopt two kinds of adversarial defenses for comparison: adversarial training methods and adversarial purification methods, especially those also based on diffusion models [69, 40, 63]. In accordance with the settings in Nie et al. [40], we conduct evaluations against strong adaptive attacks using a fixed subset of 512 randomly sampled images.

**Evaluation Metrics**  We leverage two evaluation metrics to assess the performance of our proposed defense method: *standard* accuracy and *robust* accuracy. Standard accuracy measures the performance of adversarial defenses on clean data, while robust accuracy measures the classification performance on adversarial examples generated by various attacks.

**Models**  We consider different architectures for a fair comparison with previous works. We utilize the ResNet [19] and WideResNet (WRN) [70] backbones as our base classifiers. For CIFAR10 and CIFAR100 datasets, we employ WRN-28-10 and WRN-70-16, the two most common architectures on adversarial benchmarks. For the ImageNet dataset, we select WRN-50-2, ResNet-50, and ResNet-152, which are extensively used in adversarial defenses. As for the pre-trained diffusion models, we incorporate two popular architectures, the elucidating diffusion model (EDM) [29] and the guided diffusion model [9]. The diffusion models and classifiers are trained independently on the same training dataset as in prior studies.

## 4.2  Main Results

In this section, we validate our defense method against two strong adaptive attacks: BPDA+EOT and adaptive white-box attack, i.e., PGD+EOT. The evaluation results of transfer-based attacks are deferred to Appendix B.

### 4.2.1  BPDA+EOT Attack

The combination of Backward Pass Differentiable Approximation (BPDA) [1] with Expectation over Transformation (EOT) [2] is commonly used for evaluating randomized adversarial purification methods. In order to compare with other test-time purification models, we follow the same settings in Hill et al. [20] with default hyperparameters for evaluation against BPDA+EOT attack. We conduct experiments under $\ell_\infty(\epsilon = 8/255)$ threat model on CIFAR10 and CIFAR100 datasets. Our method achieves high robust accuracies, outperforming existing diffusion-based purification methods by a significant margin, while maintaining high standard accuracies.

*CIFAR10*  Table 1 shows the robustness performance against BPDA+EOT attack under the $\ell_\infty (\epsilon = 8/255)$ threat models on CIFAR10 dataset. Mark $o$ in parentheses denotes the practi-

Table 2: Standard accuracy and robust accuracy (%) against BPDA+EOT attack under $\ell_\infty(\epsilon = 8/255)$ threat model on the CIFAR100 dataset.

| Accuracy | Hill et al. [20] | Yoon et al. [69] | Diffusion-based Purification (step) | | | | | Ours | |
| | | | 1 | 10 | 20 | 40 | 80 | (o) | (n) |
|---|---|---|---|---|---|---|---|---|---|
| Standard | 51.66 | 60.66 | 61.52 | 67.77 | 69.92 | 67.58 | 66.99 | **69.92** | **75.98** |
| Robust | 26.10 | 39.72 | 40.03 | 47.66 | 48.83 | 48.05 | 47.85 | **66.99** | **65.43** |

Table 3: Standard and robust accuracy against PGD+EOT on CIFAR-10. Left: $\ell_\infty(\epsilon = 8/255)$; Right: $\ell_2(\epsilon = 0.5)$. Compared with adversarial training (AT) and purification (AP) methods.

| Type | Method | Standard | Robust | Type | Method | Standard | Robust |
|---|---|---|---|---|---|---|---|
| WideResNet-28-10 | | | | WideResNet-28-10 | | | |
| AT | Pang et al. [42] | 88.62 | 64.95 | AT | Sehwag et al. [50] | 90.93 | 83.75 |
| | Gowal et al. [15] | 88.54 | 65.10 | | Rebuffi et al. [44] | 91.79 | 85.05 |
| | Gowal et al. [16] | 87.51 | 66.01 | | Augustin et al. [3] | 93.96 | 86.14 |
| AP | Yoon et al. [69] | 85.66 | 37.27 | AP | Yoon et al. [69] | 85.66 | 74.26 |
| | Nie et al. [40] | 90.07 | 51.25 | | Nie et al. [40] | 91.41 | 82.11 |
| | Ours | **95.02±0.30** | **67.68±0.29** | | Ours | **94.99±0.09** | **86.78±0.33** |
| WideResNet-70-16 | | | | WideResNet-70-16 | | | |
| AT | Gowal et al. [16] | 88.75 | 69.03 | AT | Rebuffi et al. [44] | 92.41 | 86.24 |
| | Wang et al. [64] | 92.97 | **72.46** | | Wang et al. [64] | **96.09** | 86.72 |
| AP | Yoon et al. [69] | 86.76 | 41.02 | AP | Yoon et al. [69] | 86.76 | 75.90 |
| | Nie et al. [40] | 90.43 | 57.03 | | Nie et al. [40] | 92.15 | 84.80 |
| | Ours | **95.18±0.09** | **71.48±0.20** | | Ours | **95.18±0.18** | **87.11±0.01** |

cal method *ScoreOpt-O* in Algorithm 1, and mark *n* denotes the method *ScoreOpt-N* in Algorithm 2. Our methods achieve surprisingly good results. Specifically, we obtain 90.59% robust accuracy, with absolute improvements of 9.19% over previous SOTA adversarial purification methods. Our method is the first adaptive test-time defense to achieve robust accuracy over 90% against the BPDA+EOT attack. Meanwhile, the standard accuracy result is on par with the best-performing method.

***CIFAR100*** We also conduct robustness evaluations against strong adaptive attacks under the $\ell_\infty$ ($\epsilon = 8/255$) threat model on the CIFAR100 dataset. The results of ScoreOpt and other adaptive purification methods are presented in Table 2. To showcase the superiority of our method over previous diffusion-based purification algorithms, we perform experiments using the approach proposed in Nie et al. [40] with varying reverse denoising steps. The optimal hyperparameter $t^*$, representing the forward timestep, is carefully tuned based on experimental results. As indicated in Table 2, increasing the reverse denoising steps only leads to marginal improvements over the one-shot denoising method. Even when the number of denoising steps is increased to 80, which will incur a large computational cost, no further improvement in robust accuracy is observed. In contrast, our ScoreOpt method achieves a notable improvement of 18.16% in robust accuracy.

### 4.2.2 Adaptive White-box Attack

In order to evaluate our method against white-box attacks, it is necessary to compute the exact gradient of the entire defense framework. However, our optimization process involves backpropagation through the U-net architecture of diffusion models, making it challenging to directly apply PGD attack. Taking inspiration from Lee and Kim [34], we approximate the full gradient using the one-shot denoising process in our experiments. The performance against PGD+EOT is shown in Table 3. Most results for baselines are taken from Lee and Kim [34]. Notably, our method significantly outperforms other diffusion-based purification methods. Specifically, compared to Nie et al. [40], our method improves robust accuracy by 16.43% under $\ell_\infty$ and by 4.67% under $\ell_2$ on WideResNet-28-10, and by 14.45% under $\ell_\infty$ and by 2.31% under $\ell_2$ on WideResNet-70-16, respectively. Compared with previous SOTA adversarial training methods, ScoreOpt achieves better robust accuracies on both WideResNet-28-10 and WideResNet-70-16 under $\ell_2$. Furthermore, under $\ell_\infty$ threat model, ScoreOpt outperforms AT baselines on WideResNet-28-10 and achieves comparable robust accuracy on WideResNet-70-16 with the top-rank model.

Table 4: Standard and robust accuracy against unseen threat models with WRN-70-16 on CIFAR10. Threat that AT is trained with is marked with an underline, and others are considered unseen.

| Defense | Standard (%) | Robust (%) | | |
| --- | --- | --- | --- | --- |
| | | $\ell_\infty$ | $\ell_2$ | StAdv |
| Base WRN-70-16 | 95.31 | - | - | - |
| Wang et al. [64] (Trained with $\ell_\infty$) | 92.97 | 72.46 | 71.68 | 3.13 |
| Wang et al. [64] (Trained with $\ell_2$) | 96.09 | 56.64 | 86.72 | 5.27 |
| Ours | 95.18 | 71.48 | 87.11 | 93.36 |

Table 5: Robustness against common corruptions on CIFAR10-C.

| Defense | gaussian | shot | impulse | elastic transform | pixelate | jpeg compression | snow | frost | fog | brightness |
| --- | --- | --- | --- | --- | --- | --- | --- | --- | --- | --- |
| Base WRN-70-16 | 25.71 | 32.19 | 27.89 | 73.17 | 44.93 | 67.32 | 77.82 | 66.49 | 71.37 | 91.33 |
| Wang et al. [64] ($\ell_\infty$) | 79.85 | 81.22 | 62.02 | 86.14 | 88.89 | 89.93 | 84.75 | 83.94 | 37.87 | 88.04 |
| Wang et al. [64] ($\ell_2$) | 81.91 | 83.17 | 63.74 | 89.96 | 92.0 | 93.61 | 89.26 | 88.17 | 48.55 | 92.88 |
| Ours | 89.94 | 87.96 | 82.68 | 83.64 | 86.36 | 89.26 | 83.5 | 85.73 | 80.2 | 91.21 |

*Approximate Gradient.* To show the effectiveness of our approximate gradients for sufficient robustness evaluation, we evaluate ScoreOpt (with only one single optimization step) on a fixed 512 CIFAR-10 subset. We use exact gradients and approximate gradients obtained by backpropagating through the one-shot denoising surrogate process, respectively. The attack success rates of PGD+EOT are 33.99% (w/ exact gradients) vs 34.11% (w/ approximate gradients) under $\ell_2$, and 50.976563% vs 50.976562% under $\ell_\infty$. Therefore, this kind of approximate gradients obtained by the one-shot denoising surrogate process can be considered as effective as computing the exact gradient to sufficiently evaluate the robustness of ScoreOpt.

### 4.3 Generalization to Different Types of Attacks

**Defense against Unseen Threats** A significant advantage of adversarial purification is that it can defend against unseen threats in a plug-and-play fashion, while most adversarial training methods exhibit robustness only against the specific threat trained with. Following Nie et al. [40], we conduct evaluations with three threat models: $\ell_\infty$, $\ell_2$, and the spatially transformed adversarial examples (StAdv). Table 4 illustrates the poor generalization of AT methods to unseen attacks and the significant generalization ability of our proposed method. The performance of AT baselines drops significantly when confronted with unseen attacks while ScoreOpt demonstrates robustness across all three threat models.

**Score-based Black-box Attack** We further assess the effectiveness of our method against the score-based black-box attack, SPSA. This evaluation is conducted on CIFAR10 with the WRN-28-10 classifier, adopting the same experimental setup as described in Yoon et al. [69], using 1,280 queries. Notably, our ScoreOpt method achieves a robust accuracy of 91.797%, in contrast to the reported 80.8% robust accuracy in Yoon et al. [69]. This demonstrates a significant performance improvement of 10% over the previous method.

**Common Corruption** To further highlight the generalization ability of ScoreOpt against various types of perturbations, we test the robustness of ScoreOpt and AT baselines on CIFAR10-C, a dataset comprising 75 frequently encountered visual distortions. The evaluation results are summarized in Table 5. where ScoreOpt consistently outperforms the AT baselines across most corruptions.

### 4.4 Further Analysis

#### 4.4.1 Combining with adversarial training methods

ScoreOpt can also be seamlessly combined with existing stronger classifiers in a plug-and-play manner to further improve robustness performance. To see this, we conduct an additional experiment by combining ScoreOpt with adversarially trained classifiers [64]. We evaluate against the PGD+EOT attack and observe improvements under both $l_\infty$ (from 71.48% to 72.57%) and $l_2$ (from 87.11% to 89.06%), respectively.

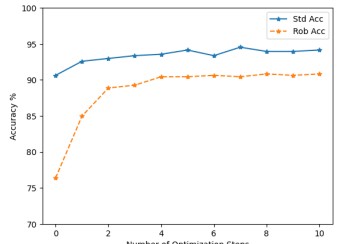

Figure 3: Accuracy with respect to optimization steps.

Table 6: The inference time cost of ScoreOpt with different optimization steps, compared with previous diffusion-based purification methods with different denoising steps.

| Denoising Steps | 1 | 5 | 10 | 20 | 50 | 100 |
|---|---|---|---|---|---|---|
| Time Cost (s) | 0.0524 | 0.1930 | 0.3271 | 0.6298 | 1.7125 | 3.3351 |
| Optimization Steps | 1 | 5 | 10 | 20 | 50 | 100 |
| Time Cost (s) | 0.1172 | 0.3632 | 0.6271 | 1.1747 | 3.2978 | 5.8181 |

### 4.4.2 Optimization Steps

Since our defense is an iterative optimization process, we conduct ablation experiments on our ScoreOpt algorithm with different optimization steps under $\ell_\infty$ ($\epsilon = 8/255$) threat model against BPDA+EOT on CIFAR10. Figure 3 shows that the standard accuracy and robust accuracy continuously increase as the number of optimization steps increases. As the number of iterations increases further, the accuracies gradually stabilize. This phenomenon shows the effectiveness of our optimization process.

### 4.4.3 Inference Speed

We compare the inference speed of ScoreOpt with diffusion purification methods that use a sequential multiple-step denoising process. We compute the inference time cost per image. As shown in Table 6, our time cost is about twice under the same steps. However, ScoreOpt needs only a few steps (about 5) to obtain significantly better results than the multi-step denoising method, which requires nearly 100 denoising steps. Therefore, compared to diffusion-based AP baselines, our method further improves the inference speed substantially in practice.

## 5 Concluding Remarks

In this work, we have proposed a novel adversarial defense framework ScoreOpt, which optimizes over the input image space to recover original images. We have introduced three optimization objectives using pre-trained score-based priors, followed by practical algorithms. We have shown that ScoreOpt can quickly purify attacked images within a few optimization steps. Experimentally, we have shown that our approach yields significant improvements over prior works in terms of both robustness performance and inference speed. We would believe our work further demonstrates the powerful capabilities of pre-trained generative models for downstream tasks.

## Acknowledgments and Disclosure of Funding

This work has been supported by the National Key Research and Development Project of China (No. 2022YFA1004002) and the National Natural Science Foundation of China (No. 12271011).

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

# A Related Work

## A.1 Adversarial attack

Adversarial attacks are aimed at manipulating or deceiving machine learning models by introducing imperceptible perturbations to input data, which can lead to misclassification. These attacks can be categorized into three types: black-box, gray-box, and white-box attacks, with increasing intensity.

In a black-box attack setting, the attacker has no knowledge of the internal structure of both the defender and the classifier. Conversely, white-box attacks have full access to all information about the pre-processor and the target classifier. Defending against white-box attacks is the most difficult task.

And gray-box attacks have partial information about the whole target model. Attackers usually have full access to the classifier but lack knowledge about the pre-processor. Adversarial examples are generated exclusively based on the classifier and evaluated using the entire model. This type of malicious perturbation is inherently limited in its effectiveness compared to white-box attacks. It is important to note that gray-box attacks can be seen as transfer-based black-box attacks, where the raw classifier serves as the source model, and the entire model (including the pre-processor) represents the target model.

## A.2 Adversarial defense

To enhance the resistance of classifiers against adversarial attacks, two primary approaches have been employed. The first approach is adversarial training (AT), which has proven to be highly effective in defending against such attacks [36, 32, 71, 44]. AT involves training classifiers using both clean data with ground-truth labels and adversarially perturbed samples. However, a notable limitation of AT is that it can only defend against attacks that the model has been specifically trained on, necessitating retraining the classifier when confronted with new attack types.

Purification, which involves applying a pre-processing procedure to input data before it is fed into classifiers, is another approach for enhancing adversarial robustness. Several existing studies have explored the effectiveness of different purification models, such as denoising auto-encoders [24, 27], sets of image filters [8], and compression-and-recovery models [26], among others. Among these purification methods, generative models [55, 12, 49] have demonstrated significant effectiveness in transforming adversarial data into clean data. The underlying idea of this technique is to learn the underlying distribution of clean data and use it to transform adversarial examples.

# B Attack Transferred from Base Classifier

We evaluate our defense against adversarial images generated from the PGD attack only on the base classifier. The attack process does not incorporate the pre-processing model. We conduct experiments against Transfer-PGD attacks for both CIFAR10 and ImageNet datasets. The experimental settings adhere to previous purification methods. In summary, our proposed method exhibits substantial improvements in both standard and robust accuracy measures.

*CIFAR10* Table 7 and 8 show the robustness results against Transfer-PGD attack under the $\ell_\infty$ ($\epsilon = 8/255$) and $\ell_2$ ($\epsilon = 0.5$) threat models on CIFAR10 dataset, respectively. We compare our method with two kinds of defenses: adversarial training and adversarial purification methods. The results demonstrate that the robust accuracies of our two methods achieve state-of-the-art performance, with comparable standard accuracies. Specifically, our method outperforms existing adversarial purification methods, including diffusion-based multi-step denoising models. Under the $\ell_\infty$ ($\epsilon = 8/255$) threat models, our proposed method improves robust accuracy by 2.2%. As for the $\ell_2$ ($\epsilon = 0.5$) threats, our algorithms still achieve a substantial improvement over existing methods in terms of both standard accuracy 93.1% and robust accuracy 92.52%.

*IMAGENET* Table 9 presents the robustness performance of our method on the Imagenet dataset. We evaluate our method under $\ell_\infty(\epsilon = 4/255)$ threat model with different perturbation budgets on three different backbones with different natural classification accuracies. Results show that the effectiveness of our method is not affected by the underlying classifier architecture. Even under larger attack budgets, we find that our method still keeps the ability to purify adversarial perturbations and generally achieves over 60% robust accuracy.

Table 7: Standard and robust accuracy against Transfer-PGD attack under $\ell_\infty(\epsilon = 8/255)$ threat model on CIFAR10, compared with other preprocessor-based adversarial defenses and adversarial training methods against transfer-based attacks.

| Type | Architecture | Method | Standard (%) | Robust (%) | Avg. (%) |
|------|-------------|--------|-------------|-----------|----------|
| Base Classifier | WRN-28-10 | - | 95.19 | 0.00 | - |
| Adversarial Training | ResNet56 | Madry et al. [36] | 87.30 | 70.20 | 78.75 |
| | | Zhang et al. [71] | 84.90 | 72.20 | 78.55 |
| Adversarial Purification | ResNet18 | Yang et al. [68] | 94.90 | 82.50 | 88.70 |
| | ResNet62 | Song et al. [55] | 90.00 | 70.00 | 80.00 |
| | | Du and Mordatch [11] | 48.70 | 37.50 | 43.10 |
| | | Grathwohl et al. [18] | 75.50 | 23.80 | 49.65 |
| | | Hill et al. [20] | 84.12 | 78.91 | 81.52 |
| | WRN-28-10 | Ho and Vasconcelos [21] | 89.26 | 80.80 | 85.03 |
| | | Yoon et al. [69] | 93.09 | 85.45 | 89.27 |
| | | Wang et al. [63] | 93.50 | 90.10 | 91.80 |
| | | Ours($o$) | 91.88±0.05 | **90.11±0.05** | 91.00 |
| | | Ours($n$) | 93.11±0.18 | **92.30±0.15** | **92.71** |

Table 8: Standard and robust accuracy against Transfer-PGD attack under $\ell_2(\epsilon = 0.5)$ threat model on CIFAR10.

| Models | Accuracy (%) | |
|--------|-------------|--------|
| | Standard | Robust |
| Base Classifier | 95.19 | 0.30 |
| Rony et al. [47] | 89.05 | 67.60 |
| Ding et al. [10] | 88.02 | 66.18 |
| Rice et al. [45] | 88.67 | 71.60 |
| Wu et al. [65] | 88.51 | 73.66 |
| Gowal et al. [15] | 90.90 | 74.50 |
| Ours($o$) | **91.85±0.02** | **90.55±0.08** |
| Ours($n$) | **93.10±0.20** | **92.52±0.14** |

Table 9: Standard and robust accuracy against Transfer-PGD $\ell_\infty$ threat model on ImageNet, under different classifier architectures and attack budgets.

| Architecture | Natural Acc | Attack Budget | Accuracy (%) | |
|-------------|------------|--------------|-------------|--------|
| | | | Standard | Robust |
| ResNet-50 | 78.52 | $\epsilon = 4/255$ | 72.85 | 70.51 |
| | | $\epsilon = 16/255$ | 69.34 | 62.70 |
| ResNet-152 | 80.66 | $\epsilon = 4/255$ | 73.83 | 71.29 |
| | | $\epsilon = 16/255$ | 72.85 | 66.60 |
| WRN-50-2 | 80.47 | $\epsilon = 4/255$ | 74.61 | 70.51 |
| | | $\epsilon = 16/255$ | 71.68 | 62.10 |

Based on the results from Table 7 to 9, we can summarize that ScoreOpt achieves better performance than previous defenses consistently and purify adversarial examples on transfer-based attacks successfully, across various datasets.

## C    Experimental Details

Code is available at `https://github.com/zzzhangboya/ScoreOpt.git`.

### C.1    Computing resources

All of our experiments are conducted using GPUs. Specifically, the diffusion models and base classifiers are trained in parallel on eight GPUs. Each test-time robustness evaluation is performed on a single GPU. The GPUs used in our experiments are NVIDIA TITAN RTX with 24GB of memory.

### C.2    Dataset descriptions

We utilize three datasets in our experiments: CIFAR-10, CIFAR-100, and ImageNet. CIFAR-10 and CIFAR-100 datasets contain 50,000 training images and 10,000 test images, with 10 and 100 classes respectively. All CIFAR images have a resolution of 32x32 pixels and three color channels (RGB). On the other hand, ImageNet consists of a validation set with 50,000 examples, featuring 1,000 classes and images with a resolution of 256x256 pixels and three color channels.

### C.3    Attack details

We conduct evaluations against the BPDA+EOT attack using 50 BPDA iteration steps and 15 EOT attack samples. Regarding the white-box PGD+EOT attack discussed in Section 4.2.2, we employ 20 PGD steps and 20 replicates for EOT attacks, consistent with the setup used in Lee and Kim [34].

For the Transfer-PGD attack, we adopt 40 PGD update iterations to generate adversarial examples, following previous studies [69, 63]. The step size $\alpha$ is set to $\epsilon/4$.

## C.4 Training details

**Diffusion models** We utilize pre-trained class-unconditional diffusion models obtained from Karras et al. [29] for CIFAR-10 and from Dhariwal and Nichol [9] for ImageNet. For the CIFAR-100 dataset, we train our own class-unconditional EDM using the ddpm++ model architecture. The training configuration and network architecture for CIFAR-100 align with the settings used in Karras et al. [29] for CIFAR-10.

**Classifiers** Our classifiers are trained alone on the training set of each dataset. For CIFAR-10, WRN-28-10 and WRN-70-16 classifiers both achieve 95.19% natural accuracy on the whole test set. For CIFAR-100, the WRN-28-10 model achieves 81.45% natural accuracy and WRN-70-16 achieves 81.28% natural accuracy.

## C.5 Implementation details of our methods

We evaluate our ScoreOpt method using the Adam optimizer in all experiments. During our experiments, we observed that both the Diff loss and the SR loss can yield satisfactory results when the number of optimization steps is relatively small. However, in comparison to SR optimization, Diff optimization offers faster inference speed and requires less memory. Unless otherwise stated, all experiments in the main text are conducted using the WRN-28-10 classifier architecture. Please refer to Table 10 for the optimization hyper-parameters we used to obtain the final evaluation results in Section 4 and Appendix B.

Table 10: Hyper-parameters choices of our optimization process for experimental results in the main text and appendix.

| Attack Type | Perturbation Budget | Method | LR | Step | Noise Level |
|---|---|---|---|---|---|
| Dataset: CIFAR10 | | | | | |
| BPDA+EOT | $\ell_\infty(\epsilon = 8/255)$ | ScoreOpt-O | 0.1 | 5 | [0.40,0.60] |
| BPDA+EOT | $\ell_\infty(\epsilon = 8/255)$ | ScoreOpt-N | 0.1 | 5 | [0.50,0.50] |
| PGD+EOT | $\ell_\infty(\epsilon = 8/255)$ | ScoreOpt-N | 0.1 | 20 | [0.25,0.25] |
| PGD+EOT | $\ell_2(\epsilon = 0.5)$ | ScoreOpt-N | 0.1 | 20 | [0.25,0.25] |
| Transfer-PGD | $\ell_\infty(\epsilon = 8/255)$ | ScoreOpt-O | 0.01 | 20 | [0.15,0.35] |
| Transfer-PGD | $\ell_\infty(\epsilon = 8/255)$ | ScoreOpt-N | 0.1 | 5 | [0.25,0.25] |
| Transfer-PGD | $\ell_2(\epsilon = 0.5)$ | ScoreOpt-O | 0.01 | 20 | [0.15,0.35] |
| Transfer-PGD | $\ell_2(\epsilon = 0.5)$ | ScoreOpt-N | 0.1 | 5 | [0.25,0.25] |
| Dataset: CIFAR100 | | | | | |
| BPDA+EOT | $\ell_\infty(\epsilon = 8/255)$ | ScoreOpt-O | 0.1 | 3 | [0.15,0.35] |
| BPDA+EOT | $\ell_\infty(\epsilon = 8/255)$ | ScoreOpt-N | 0.1 | 3 | [0.25,0.25] |

# D Ablation Studies and Additional Results

## D.1 Different classifier architectures

To showcase the effectiveness and reliability of incorporating our method into any off-the-shelf classifiers, we conduct additional evaluations using the WideResNet-70-16 classifier architecture. All of these experiments are conducted using the same optimization hyper-parameters as those employed for the WRN-28-10 architecture. Table 11 demonstrates that our method with WRN-70-16 consistently achieves high robust accuracy against various attacks on both the CIFAR-10 and CIFAR-100 datasets.

Table 11: Evaluation results obtained by WRN-70-16 classifier architecture.

| Attack Type | Perturbation Budget | Method | Acc (%) Standard | Robust |
|---|---|---|---|---|
| **Dataset: CIFAR10** | | | | |
| BPDA+EOT | $\ell_\infty(\epsilon = 8/255)$ | ScoreOpt-N | 93.75 | 91.60 |
| Transfer-PGD | $\ell_\infty(\epsilon = 8/255)$ | ScoreOpt-N | 92.47 | 92.32 |
| Transfer-PGD | $\ell_2(\epsilon = 0.5)$ | ScoreOpt-N | 92.47 | 92.29 |
| **Dataset: CIFAR100** | | | | |
| BPDA+EOT | $\ell_\infty(\epsilon = 8/255)$ | ScoreOpt-O | 76.56 | 65.82 |
| BPDA+EOT | $\ell_\infty(\epsilon = 8/255)$ | ScoreOpt-N | 69.53 | 66.99 |

## D.2 Number of EOT attack replicates

We present additional results for our ScoreOpt defense by varying the number of EOT attack replicates. All experiments are conducted using the same settings as described in Section 4, except the EOT number. Figure 4 illustrates the robustness performance against PGD+EOT attacks with different numbers of EOT replicates, while Figure 5 showcases the results against BPDA+EOT attacks. The evaluations are performed using the WRN-28-10 architecture on the CIFAR-10 dataset. Interestingly, our findings indicate that the number of EOT replicates does not significantly impact the standard and robust accuracy of our method.

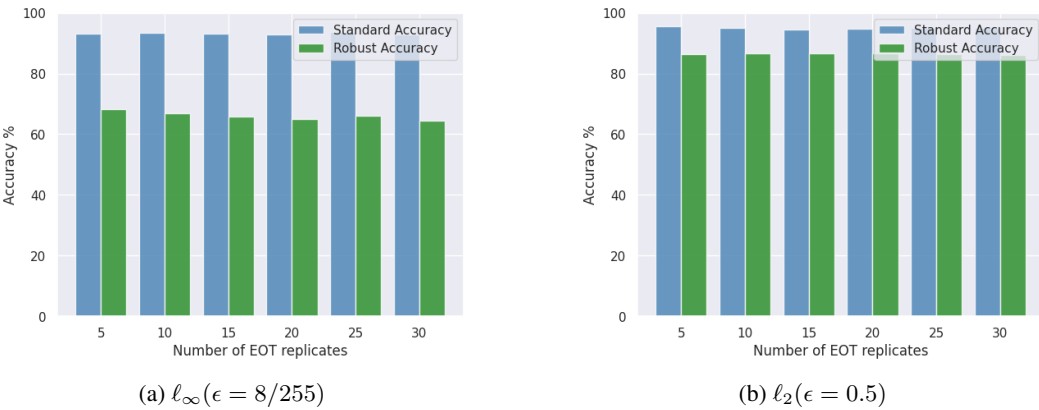

(a) $\ell_\infty(\epsilon = 8/255)$        (b) $\ell_2(\epsilon = 0.5)$

Figure 4: Robustness performance against PGD+EOT attack under the $\ell_\infty(\epsilon = 8/255)$ and $\ell_2(\epsilon = 0.5)$ threat models, respectively. We evaluate our methods with WRN-28-10 on CIFAR-10.

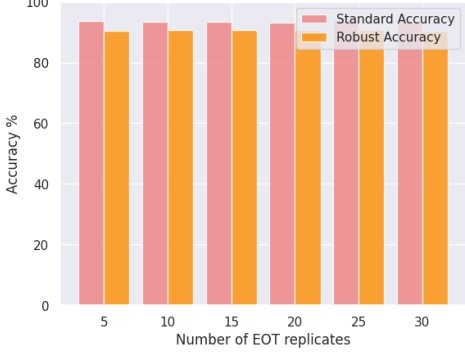

Figure 5: Robustness performance against BPDA+EOT attack under the $\ell_\infty(\epsilon = 8/255)$ threat model. We evaluate our methods with WRN-28-10 on CIFAR-10.

## D.3 Hyper-parameter choices of the optimization process

To further assess the effectiveness of our methods, we conduct additional experiments presented in Table 12. In these experiments, we use different optimization hyper-parameters compared to the ones used in Table 10. It is worth noting that varying the optimization hyper-parameters typically does not have a significant impact on the final robustness results. This observation suggests that our defense mechanism remains effective and consistent across different choices of optimization hyper-parameters, further reinforcing its reliability.

Table 12: Evaluation results with different hyper-parameters. The * symbol indicates that the hyperparameter is identical to the corresponding experiment in the main text.

| Perturbation Budget | Method | Hyper-parameter | | | Acc (%) | |
| | | LR | Step | Noise Level | Standard | Robust |
|---|---|---|---|---|---|---|
| Attack: BPDA+EOT | | | | | | |
| $\ell_\infty(\epsilon = 8/255)$ | ScoreOpt-O | * | * | [0.30,0.50] | 93.16 | 79.30 |
| $\ell_\infty(\epsilon = 8/255)$ | ScoreOpt-O | * | * | [0.30,0.60] | 92.38 | 81.05 |
| $\ell_\infty(\epsilon = 8/255)$ | ScoreOpt-N | * | * | [0.40,0.40] | 94.53 | 92.19 |
| $\ell_\infty(\epsilon = 8/255)$ | ScoreOpt-N | * | * | [0.60,0.60] | 91.60 | 89.26 |
| Attack: Transfer-PGD | | | | | | |
| $\ell_\infty(\epsilon = 8/255)$ | ScoreOpt-O | 0.1 | 10 | * | 91.42 | 88.56 |
| $\ell_2(\epsilon = 0.5)$ | ScoreOpt-O | * | * | [0.20,0.40] | 91.55 | 90.21 |
| $\ell_2(\epsilon = 0.5)$ | ScoreOpt-N | * | 10 | * | 91.64 | 91.17 |
| $\ell_2(\epsilon = 0.5)$ | ScoreOpt-N | * | 10 | [0.30,0.30] | 91.24 | 90.94 |

## D.4 More results on the whole ImageNet test set

The experimental results presented in Table 9 are evaluated on a fixed subset of 512 randomly sampled images. To provide a more comprehensive assessment, we include additional results on the complete 50,000 test set of ImageNet in Table 13.

Table 13: Standard and robust accuracy against Transfer-PGD $\ell_\infty$ threat model on Imagenet, under different classifier architectures and attack budgets.

| Architecture | Natural Accuracy | Attack Budget | Accuracy (%) | |
| | | | Standard | Robust |
|---|---|---|---|---|
| ResNet-50 | 76.15 | $\epsilon = 4/255$ | 70.07 | 66.02 |
| | | $\epsilon = 16/255$ | 66.93 | 60.45 |
| WRN-50-2 | 78.48 | $\epsilon = 4/255$ | 72.38 | 68.22 |
| | | $\epsilon = 16/255$ | 69.27 | 61.73 |

# E  Discussions and Limitations

The experiments conducted demonstrate the significant improvements achieved by our method in terms of both robustness performance and inference speed compared to previous methods. Our approach can be seamlessly integrated into various off-the-shelf classifiers. However, it is important to acknowledge a few potential drawbacks of our method.

Firstly, the computational memory cost remains a challenge that needs to be addressed. Although our method requires only a small number of optimization steps, the inclusion of the U-Net Jacobian term in the loss function introduces additional computational overhead. This memory constraint limits the evaluation of our method on stronger adversarial benchmarks like RobustBench.

Secondly, while we have experimentally demonstrated that our test-time optimization process converges to a certain local optimum within a small number of steps, we have not provided a theoretical convergence analysis.

Our future work will focus on addressing these limitations and further exploring the theoretical aspects of our method. By addressing these issues, we can enhance the practicality and theoretical understanding of our approach, paving the way for more robust and efficient adversarial defense mechanisms.

