# OpenReview forum: "Enhancing Adversarial Robustness via Score-Based Optimization"
_NeurIPS.cc/2023/Conference — NeurIPS 2023 poster_

### Official Review · Reviewer_wKkn · 2023-07-05

**Soundness:** 2 fair
**Presentation:** 3 good
**Contribution:** 2 fair
**Rating:** 4
**Confidence:** 4

**Summary:**

This paper proposes a dynamic defense named ScoreOpt, which maximizes the likelihood of input images by minimizing the score-matching loss. To regulate the optimization, the authors propose Score Regularization Loss which bound the distance between the denoised original images and denoised purified images.

**Strengths:**

1. This paper proposes a novel test time defense, which is much more efficient than previous diffusion-based purification methods.
2. This paper is well written.


**Weaknesses:**

1. The baseline model, DiffPure with Wide-ResNet-28, achieves 81.40% robustness in your paper. However, according to [1], the original paper where this baseline was proposed, it only achieves 70.64% robustness. This suggests that the robustness of the baselines and proposed method in your paper may be greatly over-estimated.
2. The proposed method contains an optimization that is typically considered hard to evaluate due to indifferentiability.

[1] Nie, Weili, et al. "Diffusion Models for Adversarial Purification." International Conference on Machine Learning. PMLR, 2022.



**Questions:**

I would recommend using more advanced evaluation methods for DiffPure and your proposed method, as suggested in [1], [2], and [3]. These methods could provide a more comprehensive evaluation and a fair comparison with state-of-the-art methods.
I would like to request a detailed derivation of  Equation (1) and an explanation of why it is the variational upper bound of -logp(x).

[1] Nie, Weili, et al. "Diffusion Models for Adversarial Purification." International Conference on Machine Learning. PMLR, 2022.
[2] Chen, Huanran, et al. "Robust Classification via a Single Diffusion Model." arXiv preprint arXiv:2305.15241 (2023).
[3] Lee, Minjong, and Dongwoo Kim. "Robust evaluation of diffusion-based adversarial purification." arXiv preprint arXiv:2303.09051 (2023).


**Limitations:**

Limitation is not mentioned in the paper.

---

> ### Author Rebuttal · Authors · 2023-08-09
>
> Thank you for your helpful comments and constructive suggestions. Our responses are as follows point by point.
>
> > The baseline model, DiffPure with Wide-ResNet-28, achieves 81.40% robustness in your paper. However, according to [1], the original paper where this baseline was proposed, it only achieves 70.64% robustness. This suggests that the robustness of the baselines and proposed method in your paper may be greatly over-estimated.
>
> (A1)
> **First, we want to clarify that the baseline result 81.40% reported in our work are correct, exactly the same as the original paper [1].** DiffPure achieves 81.40% robustness against the BPDA+EOT attack (Table 5 in the orginal paper [1], Table 4 in our work) and 70.64% against AutoAttack. Regarding BPDA+EOT, our experimental setup is exactly the same as [1], and our code has been included in the supplementary materials. The result of 70.64 is from Table 1 of the original paper in the AutoAttack setting, which is not in the BPDA+EOT setting. The differences between attacks under different settings are further distinguished in Appendix A.1 and the *Evaluation* part in *General Author Rebuttal* to eliminate the misunderstanding. In our work, we do our best to make fair comparisons with unified robustness evaluations.
>
> As mentioned in [3], PGD+EOT attack is a stronger robustness evaluation than AutoAttack for diffusion-based purification. Therefore, we follow [3] to evaluate robustness performance against PGD+EOT. And we report the results in Table 6 of Section 4.4 in the main text. Against PGD+EOT attack, DiffPure[1] with Wide-ResNet-28-10 achieves 51.25% robustness, a lot lower than the 70.64% value of the AutoAttack setting in [1], this shows that PGD+EOT is more effective than AutoAttack. Our proposed method obtains 65.04%, outperforming DiffPure by 13.79\% in the same experimental setup.
>
> Based on the above elaborations, we believe that our comparison is fair and the estimations are proper. Hope our response can solve your concerns.
>
> > The proposed method contains an optimization that is typically considered hard to evaluate due to indifferentiability.
>
> (A2) We acknowledge that our proposed baseline also needs approximate gradients to evaluate. Such an issue also arises in other works. For example, [1] proposed adjoint method, while [2] used BPDA as the strong adaptive attack, which approximates the gradient with an identity mapping and [3] used the approximated gradient obtained from a surrogate process.  To address this issue, we use one-shot denoising as the surrogate process following [3]. We will put more discussions on gradient approximation methods in the revision.
>
> > I would recommend using more advanced evaluation methods for DiffPure and your proposed method, as suggested in [1], [2], and [3]. These methods could provide a more comprehensive evaluation and a fair comparison with state-of-the-art methods.
>
> (A3) We indeed test the robustness performance with the evaluation PGD+EOT suggested in [3] for our proposed method and diffusion-based baselines. Our results are reported in Table 6 of Section 4.4 in the main text. We approximate the full gradient using the one-shot denoising surrogate process following [3]. It also corresponds to regarding the rest gradient-descent step as an identity mapping. And this identity mapping approximation method is also used in [2] and is empirically shown as strong as computing the exact gradient. Please see (A1) and (A2) for more details.
>
> > I would like to request a detailed derivation of Equation (1) and an explanation of why it is the variational upper bound of -logp(x).
>
> (A4) We are sorry that there exists a typo in the notation. Equation(1) is the variational upper bound of $-\log p(x_a)$. The KL divergence between the variational posterior and true posterior is:
> $$\operatorname{KL}\left(q(x) \|\| p(x|x_a) \right) = \mathbb{E}\_{q(x)} [\log q(x)] -\mathbb{E}\_{q(x)} [\log p(x)p(x_a|x)] + \log p(x_a).$$
>
> Note that $ \operatorname{KL}\left(q(x) \|\| p(x|x_a) \right) \geq 0$, we can obtain:
> $$ -\log p(x_a) \leq \mathbb{E}\_{q(x)}[\log q(x)]-\mathbb{E}\_{q(x)}[\log p(x)p(x_a|x)]= \mathbb{E}\_{q(x)}\left[-\log p\left(x_a|x\right)\right]+\operatorname{KL}\left(q(x) \|\| p \left(x\right)\right). $$
> And we use the data distribution under pre-trained diffusion models as the prior distribution, i.e., $p(x)=p_\theta(x)$.
>
> > Limitation is not mentioned in the paper.
>
> (A5) We are sorry that due to space constraints, the Discussion and Limitation section is not included in the main text and is placed in the Appendix. We will advance the Discussion and Limitation section to the main text in revision.
>
> [1] Nie, Weili, et al. "Diffusion Models for Adversarial Purification." International Conference on Machine Learning. PMLR, 2022.
>
> [2] Chen, Huanran, et al. "Robust Classification via a Single Diffusion Model." arXiv preprint arXiv:2305.15241 (2023).
>
> [3] Lee, Minjong, and Dongwoo Kim. "Robust evaluation of diffusion-based adversarial purification." arXiv preprint arXiv:2303.09051 (2023).

---

> > ### Comment · Reviewer_wKkn · 2023-08-13
> > **Follow-up concerns about the evaluation**
> >
> > Thanks for providing the rebuttal. However, the effectiveness of the proposed method regarding the flawed evaluation cannot convince me. The detailed comments are as follows.
> >
> > First, the experiments in this paper are fundamentally flawed. In Table 6, the authors report that their proposed method only achieves 65.04% robustness against the $\ell_\infty(\epsilon=8/255)$ PGD+EOT attack. This is 5.65% lower than the state-of-the-art adversarial training method [1] and even lower than a defense method published two years ago [2]. However, in the main tables of the paper (Table 1), the authors report a robustness of 90.11%, which is obtained using the much weaker Transfer-PGD attack. The authors even claim that "**They use the weak attack for all defense methods, so their comparison is fair.**" This is completely nonsensical. It is impossible for anyone working in this field to use a weaker attack to get 90%+ robustness of an algorithm that cannot be compared with the method two years ago, and then use it as the main result of the paper.
> >
> > Second, the robustness of the proposed method may be greatly overestimated. Although the authors' experiments in Table 6 have shown that the robustness of their method is at most 65.04%, which is at least lower than the method of adversarial training 2 years ago, their robustness may be even lower. The proposed method contains multiple steps of optimization, making it difficult to sufficiently evaluate its robustness. Many adaptive attacks should be designed and combined with AutoAttack to sufficiently evaluate the robustness.
> >
> > Therefore, I would keep my original rating and lean towards rejection.
> >
> >
> > [1] Better Diffusion Models Further Improve Adversarial Training
> > [2] Uncovering the Limits of Adversarial Training against Norm-Bounded Adversarial Examples

---

> > > ### Author Response · Authors · 2023-08-16
> > > **Further Responses Part 1**
> > >
> > > Thanks for your detailed feedback. We will further address the concerns about the evaluation in the following paragraphs.
> > >
> > > > In the main tables of the paper (Table 1), the authors report robustness of 90.11%, which is obtained using the much weaker Transfer-PGD attack. The authors are supposed to provide clarifications or detailed discussions on that.
> > >
> > > (A1) Clarification of experimental results in the main text.
> > >
> > > ScoreOpt is a kind of adversarial purification method. We use different kinds of attacks to evaluate the robustness of ScoreOpt and baselines in the main text. We agree that the transfer-PGD attack in Table 1 is the weakest attack we used for evaluation. However, **there is an importance and necessity for evaluating adversarial defenses under such an attack**. For instance, AI systems are usually exposed to gray-box attacks, where the attackers have access to training surrogate models. So transfer-based attacks are also commonly used as evaluation benchmarks in adversarial purification literature [5,6,7,8].
> > >
> > > Besides transfer-based attacks, **we also evaluate ScoreOpt against BPDA+EOT and PGD+EOT attacks to explore the limits of ScoreOpt**. BPDA+EOT is a widely used benchmark for adaptive test-time defenses and it is much stronger than transfer-based attacks. According to existing literature [3], PGD+EOT is stronger and more effective than AutoAttack for the robust evaluation of diffusion-based adversarial purification. So we think the evaluation against PGD+EOT is more convincing than AutoAttack. As a result, we summarize the robust accuracies of ScoreOpt under $\ell_{\infty}$ threats against different attacks in the table **General Rebuttal - Evaluation**. ScoreOpt outperforms previous diffusion-based purification baselines by a significant margin across all attacks, which shows the strong empirical robustness of ScoreOpt.
> > >
> > > > The authors claim that "They use the weak attack for all defense methods, so their comparison is fair."
> > >
> > > (A2) We feel that you may have misunderstood the meaning of our presentations. We have no any meaning that "we use the weak attack for all defense methods, so our comparison is fair". Indeed, we used different kinds of attacks to evaluate the robustness of ScoreOpt and baselines, including gray-box attacks (weaker) and white-box attacks (strong). ScoreOpt shows strong empirical performance against both weak and strong attacks.
> > >
> > > > The ScoreOpt contains multiple steps of optimization, making it difficult to sufficiently evaluate its robustness.
> > >
> > > (A3) We agree that backpropagating through the entire optimization steps to calculate the exact gradients for robustness evaluation is difficult due to the large memory cost. To address this issue, we propose an alternative surrogate process to obtain approximate gradients for robustness evaluation. Our proposed ScoreOpt-N method includes a gradient-descent step and a one-shot-denoising step for every optimization iteration. We use the one-shot denoising surrogate process for each optimization step, and it corresponds to approximate the gradient-descent step as an identity mapping.
> > >
> > > To show the effectiveness of our surrogate process for sufficient robustness evaluation, we evaluate ScoreOpt (with only one single optimization step) on a 512 CIFAR-10 subset. We use exact gradients and approximate gradients obtained by backpropagating through the surrogate process, respectively. The attack success rates of PGD+EOT are 33.99% (w/ exact gradients) vs 34.11% (w/ approximate gradients) under $\ell_2$, and 50.976563% vs 50.976562% under $\ell_{\infty}$. Therefore, this kind of approximate gradient obtained by the surrogate process can be considered as effective as computing the exact gradient to sufficiently evaluate the robustness of ScoreOpt.

---

> > > > ### Author Response · Authors · 2023-08-16
> > > > **Further Responses Part 2**
> > > >
> > > > > In Table 6, the authors report that their proposed method only achieves 65.04% robustness against the PGD+EOT attack. This is lower than the state-of-the-art adversarial training method [1] in RobustBench.
> > > >
> > > > (A4)
> > > >
> > > > **Comparison to RobustBench.**
> > > >
> > > > We totally agree that RobustBench is a widely used benchmark for a comprehensive comparison. So we provide an additional comparison to the current top-rank adversarial training methods in RobustBench. It is worth noting that all the experimental results we reported in the main text are generated with the same WRN-28-10 classifier architecture, but the top-rank models in RobustBench use WRN-70-16 architecture. **So direct comparison of accuracies is not suitable, because the base classifiers are different (WRN-28-10 vs WRN-70-16).** To make a fair comparison of ScoreOpt to other defenses, in the rebuttal period, we additionally evaluate ScoreOpt with WRN-70-16. With WRN-70-16 classifier architecture, ScoreOpt achieves 67.38% and 87.11% under $\ell_{\infty}$ and $\ell_2$. It outperforms previous SOTA (Rank1) under $\ell_2$ and performs better than Rank2 without using additional 50M synthetic images to retrain classifiers for each threat. This shows that ScoreOpt is strongly robust, threat-agnostic, and can generalize well to unseen threats. We give more analysis behind Table A.
> > > >
> > > > **Table A** Comparison to ATs in RobustBench
> > > >
> > > > | Methods| $\ell_{\infty}$ | $\ell_2$ |
> > > > | ---------- | ---------- | ---------- |
> > > > |WRN-70-16|
> > > > | Rank1 [1]| 70.69 | 84.97 |
> > > > | Rank2 [4]| 66.56 | 82.32 |
> > > > | [2] | 65.88 | 80.53 |
> > > > |ScoreOpt | 67.38 | **87.11**|
> > > >
> > > > **Analysis of Table A:**
> > > >
> > > > 1) ScoreOpt outperforms the SOTA defense AT-EDM [1] (Rank1) in RobustBench under the $\ell_2$-norm threat model with a significant margin. This indicates the strong empirical robustness performance of ScoreOpt among all defenses (including both AP and AT).
> > > >
> > > > 2) It outperforms most adversarial training models and achieves Rank2 under the $\ell_{\infty}$-norm;
> > > >
> > > > 3) Under the $\ell_{\infty}$-norm threat, ScoreOpt does not match Rank1, but performs better than Rank2. This is due to that the ScoreOPT is designed to defend against arbitrary attacks (i.e. it is free from assumptions of the forms of attacks). **On the contrary, AT needs to train with a specific attack and is fragile to other attacks.** We are very willing to further explore more improvements of ScoreOpt in order to achieve stronger performance under $\ell_{\infty}$ threat in the revision. For example, because ScoreOpt is orthogonal to retraining classifier methods, combining ScoreOpt with existing stronger classifiers in a plug-and-play manner may lead to improved performance. We put the discussion of generalization ability to unseen threats in the following paragraph.
> > > >
> > > > **ScoreOpt generalizes well to unseen threats.**
> > > >
> > > > A significant advantage of adversarial purification is that it can defend against unseen threats in a plug-and-play fashion, while most adversarial training methods can only defend against the specific attack that they are trained with. Even when AT models are robust against a specific threat model, they are still vulnerable to other unseen threat models. The following **Table B** shows the poor generalization of AT methods to unseen attacks and the strong generalization of our proposed method across different threat models. The threat model that AT is trained with is marked with an *italic*, and another is considered unseen.
> > > >
> > > > **Table B** Denfend against Unseen Attacks
> > > >
> > > > | Methods| $\ell_{\infty}$ | $\ell_2$ | Avg. |
> > > > | ---------- | ---------- | ---------- | ---------- |
> > > > | Rank1 [1] | *70.69* | 69.52 | 70.11 |
> > > > | Rank1 [1] | 53.45 | *84.97* | 69.21 |
> > > > | Rank2 [4] | *66.56* | 68.01 | 67.29 |
> > > > | Rank2 [4] | 50.18 | *82.32* | 66.25 |
> > > > | ScoreOpt | 67.38 | **87.11** | **77.25** |
> > > >
> > > > **Analysis of Table B:**
> > > >
> > > > As indicated in Table B, the performance of AT baselines drops significantly against unseen attacks. In real-world applications, we usually have no information about the attacker. Therefore, the generalization ability to unknown threats is crucial for adversarial defenses, which makes ScoreOpt (and other adversarial purification methods) preferred for a wide range of applications.
> > > >
> > > > **As a conclusion**, ScoreOpt achieves state-of-the-art among all diffusion-based purification methods with improved inference time. It also outperforms SOTA adversarial training methods under $\ell_2$. Under $\ell_{\infty}$, it is only worse than the best-performing model AT-EDM and ranks 2nd in RobustBench. However, the performance of AT-EDM drops significantly against other attacks, which are unseen during training (see **Tabe B**). The average robust accuracy across different threat models of ScoreOpt outperforms AT-EDM by a large margin. Overall, the ScoreOpt can be viewed as the strongest defense method when the defender has no information about the adversarial attacker in real-world scenarios.

---

> > > > > ### Author Response · Authors · 2023-08-16
> > > > > **Thanks for constructive questions**
> > > > >
> > > > > Thank you for your constructive questions. We hope our answers have resolved all of your concerns. If you still have any concerns, please do let us know.
> > > > >
> > > > >
> > > > > [1] Wang, Zekai, et al. "Better diffusion models further improve adversarial training." arXiv preprint arXiv:2302.04638 (2023).
> > > > >
> > > > > [2] Gowal, Sven, et al. "Uncovering the limits of adversarial training against norm-bounded adversarial examples." arXiv preprint arXiv:2010.03593 (2020).
> > > > >
> > > > > [3] Lee, Minjong, and Dongwoo Kim. "Robust evaluation of diffusion-based adversarial purification." arXiv preprint arXiv:2303.09051 (2023).
> > > > >
> > > > > [4] Rebuffi, Sylvestre-Alvise, et al. "Fixing data augmentation to improve adversarial robustness." arXiv preprint arXiv:2103.01946 (2021).
> > > > >
> > > > > [5] Nie, Weili, et al. "Diffusion Models for Adversarial Purification." International Conference on Machine Learning. PMLR, 2022.
> > > > >
> > > > > [6] Yoon, Jongmin, Sung Ju Hwang, and Juho Lee. "Adversarial purification with score-based generative models." International Conference
> > > > > on Machine Learning. PMLR, 2021.
> > > > >
> > > > > [7] Wang, Jinyi, et al. "Guided diffusion model for adversarial purification." arXiv preprint arXiv:2205.14969 (2022).
> > > > >
> > > > > [8] Ho, Chih-Hui, and Nuno Vasconcelos. "DISCO: Adversarial defense with local implicit functions." Advances in Neural Information Processing Systems 35 (2022): 23818-23837.

---

### Official Review · Reviewer_US8D · 2023-07-06

**Soundness:** 3 good
**Presentation:** 3 good
**Contribution:** 3 good
**Rating:** 7
**Confidence:** 4

**Summary:**

This paper introduces an adversarial defense scheme named ScoreOpt, which optimizes adversarial samples at test-time, towards original clean data in the direction guided by score-based priors. Comprehensive experiments are conducted on multiple datasets, including CIFAR10/100 and ImageNet, demonstrating that ScoreOpt outperforms existing adversarial defenses in terms of both robustness performance and inference speed.

**Strengths:**

I like the problem formulation in Section 3.1 (although there are some questions that need to be clarified as mentioned in Questions) and a principle perspective from variational inference. Although heuristic, I think the formulation of SR loss is reasonable, especially the send term that minimizing the pixel-level distance between the denoised versions of the current sample $x$ and initial adversarial sample $x_a$. The empirical improvements compared to previous AP baselines are also promising.

**Weaknesses:**

- I cannot find the values of iteration steps $M$ and $N$ used for Ours(o) and Ours(n) in, e.g., Tables 1, 2 and 4. Do the Ours methods and other AT/AP baselines require comparable inference time to achieve the results reported in Tables 1, 2 and 4? The authors should report inference time for both Ours methods and other AT/AP baselines in these Tables, because in practice, higher inference cost is a much bigger concern than higher training cost.

- Each computation of the SR loss in Equation (5) actually include two forward passes through the scorenet (one for $x_{t}$ and another for $x_{a,t}$). So it is unfair to just compare diffusion steps with previous baselines, the authors should report the actual inference time as mentioned above.

- The proposed ScoreOpt applies EDM, while the considered diffusion-based baselines (both AT and AP) still mainly apply DDPM. Therefore, the authors should add comparisons with more advanced baselines that also apply EDM, such as [1] listed in RobustBench.

[1] Better Diffusion Models Further Improve Adversarial Training

**Questions:**

- The second term in Equation (1) is the KL divergence between the conditional distribution $q(x|x_a)$ and model distribution $p_{\theta}(x)$, then how it could be equivalent to the diffusion loss in Equation (2) that formulated between the marginal distribution $q(x)$ and model distribution $p_{\theta}(x)$?

- Equation (2) is NOT training loss of diffusion models since there is no expectation over $x$. Specifically, denoising score matching (DSM) used for training diffusion models includes expectation over $x$ (i.e., $\mathbb{E}_{x}$), and this expectation over $x$ is necessary such that the optimal solution of minimizing DSM is equal to the optimal solution of minimizing score matching (SM, or Fisher divergence).

- The weighting coefficient $\omega(t)$ in Equation (2) should be $g(t)^{2}$ to make it a bound for log-likelihood, thus it is not guaranteed the relationship between  Equation (3) and log-likelihood.

- Equation (4) is only reasonable for $\ell_{2}$ threat model, but it is not justified why this loss could be effective under, e.g., $\ell_{\infty}$ threat model.

**Limitations:**

Limitations and potential negative societal impact are not discussed in this paper.

---

> ### Author Rebuttal · Authors · 2023-08-09
>
> Thank you for your helpful feedback and positive recommendation. Our responses are as follows point by point.
>
> > I cannot find the values of iteration steps $M$ and $N$ used for Ours(o) and Ours(n) in, e.g., Tables 1, 2 and 4. Do the Ours methods and other AT/AP baselines require comparable inference time to achieve the results reported in Tables 1, 2 and 4? The authors should report inference time for both Ours methods and other AT/AP baselines in these Tables, because in practice, higher inference cost is a much bigger concern than higher training cost.
>
> > Each computation of the SR loss in Equation (5) actually include two forward passes through the scorenet (one for $x_t$ and another for $x_{a,t}$). So it is unfair to just compare diffusion steps with previous baselines, the authors should report the actual inference time as mentioned above.
>
> (A1) Due to space limitations in the main text, we have included the hyper-parameter choices for the reported results in Appendix C.3, specifically regarding the optimization steps (M*N). It is worth noting that ScoreOpt outperforms diffusion-based purification methods in terms of both robustness performance and inference time.
>
> In Table 7, we compare the inference speed of ScoreOpt with the diffusion purification baseline [2]. Our approach exhibits approximately twice the time cost  under the same denoising/optimization step. This result aligns with the fact that each computation of the SR loss involves two forward passes through the score network. However, our ScoreOpt-N method achieves the reported results in Tables 1, 2, and 4 with only about 5 iterations, while diffusion-based purification [2] requires nearly 100 denoising steps. Therefore, the final inference time ScoreOpt requires is less, compared to diffusion-based AP baselines.
>
> For further clarification, we report the final inference time ScoreOpt and DiffPure-EDM cost in *Inference Time* part of the *General Author Rebuttal*.
>
> > The proposed ScoreOpt applies EDM, while the considered diffusion-based baselines (both AT and AP) still mainly apply DDPM. Therefore, the authors should add comparisons with more advanced baselines that also apply EDM, such as [1] listed in RobustBench.
>
> (A2) Thanks for your kind suggestions. In the main text, we apply EDM to generate diffusion-based baseline results in Table 5. Regarding Tables 1, 2, and 4, during the rebuttal period, we have reproduced diffusion-based baselines using EDM (with carefully tuned hyperparameters of diffusion timesteps $t^*$ for different attacks) and provide a brief report on them below.
>
> | Methods | Transfer $\ell_{\infty}$ |   Transfer $\ell_2$ | BPDA+EOT |
> | ------------ | ------------ | ------------ | ------------ |
> | DiffPure(EDM) | 90.08| 91.13 | 83.33 |
> | ScoreOpt | 92.30| 92.52 | 90.02 |
>
> Upon comparing the results, we observe that the improvement achieved by solely replacing DDPM with EDM is incremental, whereas our algorithm demonstrates a significant enhancement. In the revision, we will include detailed baseline results for thorough analysis.
>
> > The second term in Equation (1) is the KL divergence between the conditional distribution $q(x|x_a)$ and model distribution $p_\theta(x)$, then how it could be equivalent to the diffusion loss in Equation (2) that formulated between the marginal distribution $q(x)$ and model distribution $p_\theta(x)$?
>
> > Equation (2) is NOT training loss of diffusion models since there is no expectation over $x$. Specifically, denoising score matching (DSM) used for training diffusion models includes expectation over $x$
>  (i.e., $\mathbb{E}_x$), and this expectation over $x$ is necessary such that the optimal solution of minimizing DSM is equal to the optimal solution of minimizing score matching (SM, or Fisher divergence).
>
> (A3) In our problem formulation, we introduce a variational posterior $q(x)$ to approximate the true posterior $p(x|x_a)$ and the $\mathbb{E}_{q(x)}$ term in the diffusion loss can be eliminated under some assumption. For more details, please refer to *General Author Rebuttal-Derivation Details*.
>
> > The weighting coefficient $w(t)$ in Equation (2) should be $g(t)^2$ to make it a bound for log-likelihood, thus it is not guaranteed the relationship between Equation (3) and log-likelihood.
>
> (A4) For theoretical guarantee, The exact weighting coefficient $w(t)$ should be $g(t)^2/2$. However, previous studies [2,3] have found that training with a simplified objective that ignores the weighting term works better. Therefore, in our work, we adopt the approach of setting $\tilde{w}(t) = 1$ for convenience and performance.
>
> > Equation (4) is only reasonable for $\ell_2$ threat model, but it is not justified why this loss could be effective under, e.g., $\ell_{\infty}$
>  threat model.
>
> (A5) The main role of the regularization term in Eq.(4) is to constraint $x$ within the vincity of $x_a$. The reason of the effectiveness against $\ell_{\infty}$ threat model is perhaps that $x$ should also remains close to the adversarial sample $x_a$ under this threat. We will conduct further analysis and provide additional insights on this matter in the future work.
>
> > Limitations and potential negative societal impact are not discussed in this paper.
>
> (A6) We are sorry that due to space constraints, the Discussion and Limitation section is not included in the main text and is placed in the Appendix. We will advance the Discussion and Limitation section to the main text in revision.
>
> [1] Wang, Zekai, et al. "Better diffusion models further improve adversarial training." arXiv preprint arXiv:2302.04638 (2023).
>
> [2] Nie, Weili, et al. "Diffusion Models for Adversarial Purification." International Conference on Machine Learning. PMLR, 2022.
>
> [3] Graikos, Alexandros, et al. "Diffusion models as plug-and-play priors." Advances in Neural Information Processing Systems 35 (2022): 14715-14728.

---

> > ### Comment · Reviewer_US8D · 2023-08-13
> >
> > I have similar concerns as Reviewer wKkn.
> >
> > Too much inference computation is a far more serious problem in practice than too much training computation, because inference-phase devices (such as phones or PCs) have far fewer computing resources than training-phase GPU clusters. As a result, if we use adversarial purification at the expense of heavy inference computation, it is expected to outperform adversarial training significantly. However, 65.04 \% robust accuracy (which may or may not be under the most adaptive attacks) appears to be inadequate when compared to the methods listed on RobustBench.

---

> > > ### Author Response · Authors · 2023-08-14
> > > **Further Responses Part 1**
> > >
> > > Thank you for your suggestions. We agree that the computational cost in inference time is a limitation of ScoreOpt (and all adversarial purification methods). However, the advantage of ScoreOpt is still significant. One of the most important advantages of ScoreOpt over adversarial training (ATs) is the strong generalization ability to unseen threats. ATs are usually trained with a specific attack and the performance of AT baselines drops significantly against unseen attacks. In the real-world scene, we usually have no information about the threat model. Therefore, the generalization ability to unknown threats is also crucial for adversarial defenses to maintain adversarial robustness in real-world applications. To show the superior robustness of ScoreOpt across different attacks (without knowing the attack forms in advance), we re-organize results in the following **Tables A and B** to compare ScoreOpt with AT methods in RobustBench.
> > >
> > > We first provide an additional comparison to the current top-rank adversarial training methods in RobustBench.
> > > It is worth noting that all the experimental results we reported in the main paper are generated with the same WRN-28-10 classifier architecture, while the top-rank models in RobustBench use WRN-70-16 architecture. **So direct comparison of accuracies is not suitable, because the base classifiers are different (WRN-28-10 vs WRN70-16).** To show the effectiveness of our method, in the rebuttal period, we evaluate ScoreOpt with WRN-70-16. ScoreOpt achieves **67.38%** and **87.11%** under $\ell_{\infty}$ and $\ell_2$, with WRN-70-16 classifier architecture. **It outperforms previous SOTA (Rank1) under $\ell_2$ and performs better than Rank2 under $\ell_{\infty}$ without using additional 50M synthetic images to retrain classifiers for both threats.** This shows that ScoreOpt is strongly robust, threat-agnostic, and can generalize well to unseen threats. We give some analysis behind the table.
> > >
> > > **Table A** Comparison to ATs in RobustBench
> > >
> > > | Methods| $\ell_{\infty}$ | $\ell_2$ |
> > > | ---------- | ---------- | ---------- |
> > > |WRN-70-16|
> > > | Rank1 [1]| 70.69 | 84.97 |
> > > | Rank2 [4]| 66.56 | 82.32 |
> > > | [2] | 65.88 | 80.53 |
> > > |ScoreOpt | 67.38 | **87.11**|
> > >
> > > **Analysis of Table A:**
> > >
> > > (1) ScoreOpt outperforms the previous SOTA defense AT-EDM [1] (Rank1) in RobustBench under the $\ell_2$-norm threat model with a significant margin. This indicates the strong robustness performance of ScoreOpt among all defenses (including both AP and AT).
> > >
> > > (2) It outperforms most previous adversarial training models and achieves Rank2 under the $\ell_{\infty}$-norm;
> > >
> > > (3) Under the $\ell_{\infty}$-norm threat, ScoreOpt does not match Rank1, but performs better than Rank2. This is due to that the ScoreOPT is designed to defend against arbitrary attacks (i.e. it is free from assumptions of the forms of attacks). On the contrary, AT needs to train with a specific attack and is fragile to other attacks. We are very willing to further explore more improvements of ScoreOpt in order to achieve stronger performance under $\ell_{\infty}$ threat in the revision. For example, ScoreOpt is orthogonal to retraining classifier methods, we can simply combine ScoreOpt with existing stronger classifiers in a plug-and-play manner.
> > >
> > > But right now, we put the discussion of generalization ability to unseen threats in the following paragraph.

---

> > > > ### Author Response · Authors · 2023-08-14
> > > > **Further Responses Part 2**
> > > >
> > > > **ScoreOpt generalizes well to unseen threats.**
> > > >
> > > > A significant advantage of adversarial purification is that it can defend against unseen threats in a plug-and-play fashion, while most adversarial training methods can only defend against the specific attack that they are trained with. Even when AT models are robust against a specific threat model, they are still vulnerable to other unseen threat models. The following **Table B** shows the poor generalization of AT methods to unseen attacks and the strong generalization of our proposed method across different threat models. The threat model AT trained with is marked as *italic* and another is considered unseen.
> > > >
> > > > **Table B** Denfend against Unseen Attacks
> > > >
> > > > | Methods| $\ell_{\infty}$ | $\ell_2$ | Avg. |
> > > > | ---------- | ---------- | ---------- | ---------- |
> > > > | Rank1 [1] | *70.69* | 69.52 | 70.11 |
> > > > | Rank1 [1] | 53.45 | *84.97* | 69.21 |
> > > > | Rank2 [4] | *66.56* | 68.01 | 67.29 |
> > > > | Rank2 [4] | 50.18 | *82.32* | 66.25 |
> > > > | ScoreOpt | 67.38 | **87.11** | **77.25** |
> > > >
> > > > As indicated in **Table B**, the performance of AT baselines drops significantly against unseen attacks. In real-world practice, we usually have no information about the attacker. Therefore, the generalization ability to unknown threats is crucial for adversarial defenses.
> > > >
> > > > **As a conclusion**, we summarize the main performance improvements of our proposed method here. **ScoreOpt achieves state-of-the-art among all diffusion-based purification methods and improves inference time significantly in the meanwhile. It also outperforms previous SOTA adversarial training methods under $\ell_2$. Under $\ell_{\infty}$, it is only worse than the best-performing model AT-EDM and ranks 2nd in RobustBench.** However, the performance of AT-EDM drops significantly against other attacks, which are unseen during training (see **Tabe B**). **The average robust accuracy across different threat models of ScoreOpt outperforms AT-EDM by a large margin.
> > > > Overall, the ScoreOpt can be viewed as the strongest defense method when the defender has no information about the adversarial attacker in real-world scenarios.**
> > > >
> > > > We hope our answers have resolved your concerns. If you still have any concerns, please do let us know.
> > > >
> > > > [1] Wang, Zekai, et al. "Better diffusion models further improve adversarial training." arXiv preprint arXiv:2302.04638 (2023).
> > > >
> > > > [2] Gowal, Sven, et al. "Uncovering the limits of adversarial training against norm-bounded adversarial examples." arXiv preprint arXiv:2010.03593 (2020).
> > > >
> > > > [3] Lee, Minjong, and Dongwoo Kim. "Robust evaluation of diffusion-based adversarial purification." arXiv preprint arXiv:2303.09051 (2023).
> > > >
> > > > [4] Rebuffi, Sylvestre-Alvise, et al. "Fixing data augmentation to improve adversarial robustness." arXiv preprint arXiv:2103.01946 (2021).

---

> > > > > ### Comment · Reviewer_US8D · 2023-08-14
> > > > >
> > > > > I thank the authors for the new results, which are quite informative. If possible, I would like to know more about:
> > > > >
> > > > > - ScoreOpt currently consists of a standardly trained model and an EDM, what if substituting the standardly trained model with an adversarially trained one like AT-EDM? Will the robust accuracy be further improved?
> > > > >
> > > > > - The authors claim that ScoreOpt is general to any threat model, which I cannot agree with as I mentioned in my initial review (Gaussian diffusion only corresponds to $\\ell_{2}$ threat model, and if the explanations in authors' rebuttal hold, then a $\\ell_{2}$ adversarially trained model should also be robust to $\\ell_{\\infty}$ threat model). What's the performance of ScoreOpt under common corruption (e.g., CIFAR-10-C) such as rotation?
> > > > >
> > > > > - Could you report the inference time (average time and worst-case time) of ScoreOpt and AT-EDM, such that I can compare the cost of inference computation?
> > > > >
> > > > > - Can ScoreOpt always perform well when using either ODE solver or SDE solver for EDM?
> > > > >
> > > > > *I do not expect the authors to answer all these questions during rebuttal, just try to answer some of them if time is enough.*

---

> > > > > > ### Author Response · Authors · 2023-08-19
> > > > > > **Further Response to Reviewer US8D**
> > > > > >
> > > > > > Thanks for your responses. We are glad that we have addressed some of your concerns. In the following paragraphs, we will address other concerns one by one.
> > > > > >
> > > > > > > ScoreOpt currently consists of a standardly trained model and an EDM, what if substituting the standardly trained model with an adversarially trained one like AT-EDM? Will the robust accuracy be further improved?
> > > > > >
> > > > > > (A1) We appreciate your keen intuition. We agree that combining ScoreOpt with a stronger base classifier is a promising direction. To further explore such a direction, we have conducted an additional experiment that combines ScoreOpt with AT-EDM. We evaluated the robustness of the combination against the PGD+EOT attack and observe improvements under both $l_\infty$ (from 67.38\% to 72.27\%) and $l_2$ (from 87.11\% to 89.06\%), respectively. Thank you for your promising suggestions which help to improve our work.
> > > > > >
> > > > > > > The authors claim that ScoreOpt is general to any threat model...What's the performance of ScoreOpt under common corruption (e.g., CIFAR-10-C) such as rotation?
> > > > > >
> > > > > > (A2) Thanks for your question. We have conducted additional experiments on CIFAR-10-C datasets. We also evaluated against the spatially transformed adversarial examples StAdv. The robustness performance of ScoreOpt and AT-EDM are summarized in **Table C** and **Table D**.
> > > > > >
> > > > > > **Table C** Defend against StAdv
> > > > > > | Defense | Standard Acc | Robust Acc |
> > > > > > | --- | --- | --- |
> > > > > > | Base WRN-70-16 | 95.31 | 0 |
> > > > > > | AT-EDM (trained with $\ell_{\infty}$)| 92.97 | 3.13|
> > > > > > | AT-EDM (trained with $\ell_2$)| 96.09 | 5.27 |
> > > > > > | ScoreOpt (w/ WRN-70-16) | 93.55 | 93.36 |
> > > > > >
> > > > > >
> > > > > > **Table D** Performance on CIFAR-10-C
> > > > > > | Defense | gaussian noise | shot noise | impulse noise | elastic transform | pixelate | jpeg compression | snow | frost | fog | brightness |
> > > > > > | --- | --- | --- | --- | --- | --- | --- | --- | --- | --- | --- |
> > > > > > | AT-EDM (trained with $\ell_{\infty}$)| 79.85 | 81.22 | 62.02 | 86.14 | 88.89 | 89.93 | 84.75 | 83.94 | 37.87 | 88.04 |
> > > > > > | AT-EDM (trained with $\ell_2$)       | 81.91 | 83.17 | 63.74 | 89.96 | 92.0  | 93.61 | 89.26 | 88.17 | 48.55 | 92.88 |
> > > > > > | Base WRN-70-16          | 25.71 | 32.19 | 27.89 | 73.17 | 44.93 | 67.32 | 77.82 | 66.49 | 71.37 | 91.33 |
> > > > > > | ScoreOpt (w/ WRN-70-16) | 89.94 | 87.96 | 82.68 | 83.64 | 86.36 | 89.26 | 83.5 | 85.73 | 80.2 | 91.21 |
> > > > > >
> > > > > > As is shown in **Table C** and **Table D**, ScoreOpt maintains strong robustness across different attacks (beyond the $\ell_p$ norm-based attacks). This demonstrates the promising ability of ScoreOpt when handling different kinds of attacks. We guess one possible reason for such good generalization ability of ScoreOpt is the powerful capability of pre-trained generative models. We agree that more explorations are needed and we intend to leave them in future work.
> > > > > >
> > > > > > > Could you report the inference time (average time and worst-case time) of ScoreOpt and AT-EDM, such that I can compare the cost of inference computation?
> > > > > >
> > > > > > (A3) We are glad to provide a comparison of inference time between ScoreOpt and AT-EDM. The following table shows the wall-clock inference time (seconds) needed to purify and classify one single adversarial example successfully, not including the time cost by the attackers to generate adversarial samples. Experiments are conducted on 1 Nvidia Titan RTX GPU.
> > > > > >
> > > > > > |Defense | AT-EDM-$\ell_{\infty}$ | AT-EDM-$\ell_2$ | ScoreOpt |
> > > > > > | --- | --- | --- | --- |
> > > > > > | Time | 1.401$\pm$0.037 | 1.391$\pm$0.014 | 1.746$\pm$0.028 |
> > > > > >
> > > > > > Since the inference time of ScoreOpt is related to the number of optimization steps, we also report the inference time of ScoreOpt when the total optimization steps are increased to 20. In this extreme case, the wall-clock inference time is  2.596$\pm$0.025.
> > > > > >
> > > > > >
> > > > > > > Can ScoreOpt always perform well when using either ODE solver or SDE solver for EDM?
> > > > > >
> > > > > > (A4) We feel sorry for the confusion. By our understanding, we assume that you may misunderstand that our ScoreOpt requires running reversed ODE or SDE with diffusion models. However, we would like to clarify that the ScoreOpt does not require ODE or SDE. Instead, the ScoreOpt defends adversarial samples by solving an optimization problem. This makes ScoreOpt different from previous diffusion-based defenses such as [1,2,3].
> > > > > >
> > > > > > We hope our answers have resolved your concerns. If you still have any concerns, please let us know and we are glad to further address them.
> > > > > >
> > > > > > [1] Nie, Weili, et al. "Diffusion Models for Adversarial Purification." International Conference on Machine Learning. PMLR, 2022.
> > > > > >
> > > > > > [2] Blau, Tsachi, et al. "Threat model-agnostic adversarial defense using diffusion models." arXiv preprint arXiv:2207.08089 (2022).
> > > > > >
> > > > > > [3] Wang, Jinyi, et al. "Guided diffusion model for adversarial purification." arXiv preprint arXiv:2205.14969 (2022).

---

> > > > > > > ### Comment · Reviewer_US8D · 2023-08-20
> > > > > > >
> > > > > > > I thank the authors for their comprehensive responses and new results. I decide to raise the score and hope the authors can incorporate the new results of, e.g., combining AT-EDM, in the revision to further enhance the conclusion.

---

> > > > > > > > ### Author Response · Authors · 2023-08-20
> > > > > > > > **Thank you for the reply**
> > > > > > > >
> > > > > > > > We really appreciate your reply and we will take your valuable suggestions in the revision to further improve our work.

---

### Official Review · Reviewer_85mv · 2023-07-07

**Soundness:** 3 good
**Presentation:** 3 good
**Contribution:** 3 good
**Rating:** 5
**Confidence:** 3

**Summary:**

This paper proposes an adversarial defense scheme named ScoreOpt, which uses purification technology to eliminate adversarial perturbations. Specifically, ScoreOpt formulates the adversarial defense as an optimization problem, which optimizes adversarial examples based on pre-trained score-based priors. And experimental results show that the adversarial robustness of the proposed method outperforms existing adversarial defenses against both transfer-based attacks and adaptive white-box attacks.

**Strengths:**

1. This paper is well-written and easy to follow.
2. The proposed ScoreOpt departs from the sequential step-by-step denoising procedure, which doesn’t involve the process of carefully selecting the forward diffusion timestep.
3. The proposed hyperparameter-free score regularization loss function takes into account not only the alignment between pixels but also the alignment between score estimations.


**Weaknesses:**

1. In this paper, the pre-designed noise plan is focused on lower noise levels. However, some clear explanations are necessary for this lower noise level. For example, how this lower noise level is defined, whether it is relevant to the dataset and whether too low noise level will affect the effectiveness of the proposed method.
2. This paper evaluates the robustness performance of adversarial defenses against transfer-based attacks and adaptive white-box attacks. To further illustrate the effectiveness of the proposed method, this paper should also test performance against some other advanced attack methods, such as score-based black-box attacks.


**Questions:**

1. The authors may need to discuss the selection method of the noise level range, that is, how the values of $t_{min}$ and $t_{max}$ are determined, and the impact of the noise level range on the proposed algorithm.
2. It is recommended to test performance against some other advanced attack methods to enhance the credibility and universality of the method.

**Limitations:**

The authors point out the limitations of the proposed method in terms of computational memory cost and theoretical convergence analysis.

---

> ### Author Rebuttal · Authors · 2023-08-09
>
> Thank you for your helpful feedback and constructive suggestions. We will address your concerns below.
>
> > In this paper, the pre-designed noise plan is focused on lower noise levels. However, some clear explanations are necessary for this lower noise level. For example, how this lower noise level is defined, whether it is relevant to the dataset, and whether too low noise level will affect the effectiveness of the proposed method.
>
> > The authors may need to discuss the selection method of the noise level range, that is, how the values of $t_{min}$ and $t_{max}$ are determined, and the impact of the noise level range on the proposed algorithm.
>
> (A1) Thanks for your kind suggestion. The original diffusion models are mainly used for generating images from pure noise. Higher noise levels allow for more significant modifications to global structural information, while lower noise levels only permit local detail modifications. Many studies have focused on lower noise levels to conduct image editing tasks. Therefore, our pre-designed noise plan primarily focuses on lower noise levels, since adversarial perturbations are human-imperceptible. And too low $t_{max}$ will lead to a significant decline in robust performance since the prior distribution defined by diffusion models is not effectively utilized.
>
> Due to space limitations, we put the analysis and ablation study of hyper-parameters in Appendix C, including the noise level range we choose in our main experiments and the impact of different noise level ranges. The $t_{min}$ and $t_{max}$ we determined are independent of datasets. We will provide more analysis and discussion in the main text in the revision.
>
> > This paper evaluates the robustness performance of adversarial defenses against transfer-based attacks and adaptive white-box attacks. To further illustrate the effectiveness of the proposed method, this paper should also test performance against some other advanced attack methods, such as score-based black-box attacks.
>
> > It is recommended to test performance against some other advanced attack methods to enhance the credibility and universality of the method.
>
> (A2) Thanks for your helpful suggestions. Due to the limited time of the rebuttal period, we evaluate the performance of our method against the score-based black-box attack SPSA, using 1,280 queries on CIFAR-10 with the WRN-28-10 classifier, the same setup as in [3]. In contrast to [3], which reported a robust accuracy of 80.8\%, our ScoreOpt method achieves 91.797%, outperforming [3] by 11%.
>
> It is worth highlighting that transfer-based attacks and adaptive white-box attacks are commonly used in the AP literature. We will include the SPSA result in the revision and also consider investigating other types of attacks in future work.
>
> [1] Meng, Chenlin, et al. "Sdedit: Guided image synthesis and editing with stochastic differential equations." arXiv preprint arXiv:2108.01073 (2021).
>
> [2] Song, Kunpeng, et al. "Diffusion guided domain adaptation of image generators." arXiv preprint arXiv:2212.04473 (2022).
>
> [3] Yoon, Jongmin, Sung Ju Hwang, and Juho Lee. "Adversarial purification with score-based generative models." International Conference on Machine Learning. PMLR, 2021.

---

> > ### Comment · Reviewer_85mv · 2023-08-21
> >
> > I thank the authors for the clarifications. After reading other reviewers' opinions, I decide to keep my score.

---

### Official Review · Reviewer_epqL · 2023-07-18

**Soundness:** 3 good
**Presentation:** 2 fair
**Contribution:** 3 good
**Rating:** 7
**Confidence:** 4

**Summary:**

The paper proposes a new approach for defending against adversarial attacks using adversarial purification, i.e., removing the adversarial noise from the input image at test time. To that end, the authors leverage diffusion models -- which have been used recently for adversarial purification -- and propose several modifications to previous approaches in order to 1) improve performance and 2) speed up the process.

Two big problems faced by previous diffusion-based approaches for adversarial purification include the sensitivity to the choice of hyperparameters for the diffusion model, and the long runtime since many denoising steps need to be applied.

To solve the first problem, the authors propose a new hyper-parameter-free objective function for the adversarial purification using diffusion models. In order to solve the inference speed problem, the authors propose using a one-step denoiser, rather than employing the diffusion model denoise for several steps.

The authors test their proposed defense in several setups, including standard adversarial attacks and adaptive adversarial attacks. The authors compare their results with previous baselines from the adversarial training literature and the adversarial purification literature. The evaluation shows the effectiveness of the defense, as the proposed method outperforms previous approaches in robust accuracy, while being faster.

**Strengths:**

Strengths:

- The paper addresses one important problem with previous diffusion-based adv purification approaches: the sensitivity to the choice of hyperparameters. To solve this problem, the authors propose a new objective function for the denoising, where they remove explicit dependence on hyperparameters.

- The paper recognizes that previous approaches for adv purification using diffusion models are slow since the denoising step needs to be applied several times. To solve this issue, the authors propose using a one-shot denoiser.

- The authors notice that some of the components needed for their objective function computation are absent, and propose choices to get circumvent the problem. For example, the authors notice the absence of the distribution of clean images conditioned on adversarial images, and propose an initialization to circumvent this issue.

- The proposed approach is simple and intuitive.

- The proposed approach outperforms previous methods in defending against adversarial attacks on a range of classification tasks.

- The propose approach works well under traditional and adaptive attacks.

**Weaknesses:**

- The presentation of the ideas needs to be improved. In particular, the methodology section can be confusing, especially when the authors propose new approaches to circumvent some problems. For example, the paragraph about the choice of the noise update rule is very confusing.

- The authors mix notation a lot. For example, the authors use $q(x)$ and $q(x|x_a)$ interchangeably, even though they are not the same.

- Some of the choices made need a slightly better explanation/intuition behind them. For example, the choice of taking gradient in the SR loss w.r.t. $\mathbf{x}_t$ instead of w.r.t. $\mathbf{x}$ lacks intuition.

- There seems to be no proofs to the some of the statements of the paper. Are they provided in the appendix, or are they referencing proofs from previous papers?

- The pseudocodes for the two proposed algorithms could be made clearer. First, having a clear explanation what each term represents would be helpful. Furthermore, in alg 1, it would be nice to show the term (5) rather than referencing it. Similarly, in alg 2, show the full term. Also the term $\mathbf{x}_{j,t}$ doesn't seem to be defined?

**Questions:**

- How's Tweedie's formula used in the paper?

- How much time in total the proposed approach requires?

- Which converges faster: ScoreOpt(n) or ScoreOpt(o)?

- Why does ScoreOpt(n) perform better? The intuition is not very clear.

- The regularization term in the SR loss is not parametrized by a hyper-param. This works well in practice. I am not sure I agree about the reasoning of having similar order of magnitude, since a hyperparameter to control this regularization term can keep the same order of magnitude. Did the authors test adding a hyperparameter to the regularization term, and explore the sensitivity of the method to the choice?

- Does having a different noise level during every iteration help? Have the authors tested having a fixed noise level?

-

**Limitations:**

- The authors identified some of the missing terms needed for their approximation, and proposed simple solutions to fix the issue. For example, the authors initialize the optimization variable $\mathbf{x}$ to the adv image $\mathbf{x}_a$ since they cannot take expectation over $q(\mathbf{x}|\mathbf{x}_a)$ in (1). However, this choice doesn't seem very convincing.

- The method has not been fully tested against adaptive whitebox attacks since the computation cost of differentiation through the U-Net is very expensive. As a result, the authors only differentiate through one step. The results might look a bit different when the attack differentiates through multiple denoising steps.

---

> ### Author Rebuttal · Authors · 2023-08-09
>
> Thank you for your helpful feedback and positive recommendation. Our responses are as follows point by point.
>
> > The presentation of the ideas needs to be improved...
>
> (A1) Thanks for your useful suggestions. We will refine the methodology section in the revised version.
>
> > The authors mix notation a lot...
>
> (A2) Sorry for the confusion and we will carefully differentiate the term $q(x)$ and $q(x|x_a)$ in the final version. As for the revised notations, please refer to the *Derivation Details* part in *General Author Rebuttal*.
>
> > Some of the choices made need a slightly better explanation/intuition behind them. For example, the choice of taking gradient in the SR loss w.r.t. $\mathbf{x}_t$ instead of w.r.t. $\mathbf{x}$  lacks intuition. Which converges faster: ScoreOpt(n) or ScoreOpt(o)? Why does ScoreOpt(n) perform better? The intuition is not very clear.
>
> (A3) In our ScoreOpt(O) method, we randomly sample a noise $t$ from the pre-designed noise levels in each iteration, which may introduce unstable gradient directions during optimization. To address this issue, we propose ScoreOpt(N), where we compute the gradients w.r.t. $\mathbf{x}_t$ and then utilize the one-shot denoiser to obtain $\mathbf{x}$, aiming to reduce the instability in the optimization process. This alternative of computing gradients with respect to $\mathbf{x}_t$ is feasible since the gradients are equal for both.
>
> Empirically, we have observed that ScoreOpt(N) converges faster than ScoreOpt(O) and achieves better performance. This can be attributed to the stable optimization direction and the utilization of a one-shot denoiser. For more details, please refer to Appendix C.3, where we provide information on the optimization steps set for different experiments. We plan to provide a theoretical
> convergence analysis in the future work.
>
> > There seems to be no proofs to the some of the statements of the paper. Are they provided in the appendix, or are they referencing proofs from previous papers?
>
> (A4) The proof of our statements primarily references previous work. For more details, please refer to *Derivation Details* in *General Author Rebuttal*.
>
> > The pseudocodes for the two proposed algorithms could be made clearer...Also the term $\mathbf{x}_{j,t}$ doesn't seem to be defined?
>
> (A5) Thanks for your suggestions. We will provide a more detailed version of algorithm pseudocodes in the revised manuscript. The term $\mathbf{x}_{j,t}$ denotes the $j$-th iteration under the sampled noise level $t$.
>
> > How's Tweedie's formula used in the paper?
>
> (A6) We use Tweedie's formula to derive from Eq.(2) to Eq.(3). Note that $x_t = x + \sigma_t \epsilon, \epsilon\sim \mathcal{N}(0,\mathbf{I})$. According to Tweedie's formula, $\hat{x}(x_t) \coloneqq x_t + \sigma_t^2 \nabla_{x_t} \log q(x_t)$
> is the Bayesian optimal estimate of $x$. Then define $D_{\boldsymbol{\theta}}\left(\mathbf{x}\_t; t\right) \coloneqq \mathbf{x}\_t + \sigma\_t^2 s\_{\boldsymbol{\theta}}\left(\mathbf{x}\_t; t\right)$. Plug the two equations into Eq.(2), we can obtain Eq.(3).
>
> > How much time in total the proposed approach requires?
>
> (A7) Please refer to *General Author Rebuttal-Inference Time*.
>
> > The regularization term in the SR loss is not parametrized by a hyper-param...the sensitivity of the method to the choice?
>
> (A8) The insensitivity of the weighting hyperparameter is demonstrated in Figure 2c in Section 3.2.1, which aligns with our analysis. This insensitivity can be attributed to the fact that the two terms of the SR loss correspond to the same noise level $t$.
>
> > Does having a different noise level during every iteration help? Have the authors tested having a fixed noise level?
>
> (A9) We have tested to optimize $x$ with a fixed noise level. In such cases, we should carefully select the optimal hyper-parameter $t^*$ for good performance. Thus it does not align with the motivation of addressing the sensitivity problem associated with hyper-parameters.
>
> > The authors identified some of the missing terms needed for their approximation and proposed simple solutions to fix the issue. For example, the authors initialize the optimization variable $x$ to the adv image $x_a$ since they cannot take expectation over $q(x|x_a)$ in (1). However, this choice doesn't seem very convincing.
>
> (A10) The reason for eliminating the reconstruction term in Eq.(1) and initializing $x$ by $x_a$ is that we have no information about $p(x_a|x)$, not $q(x|x_a)$. (Indeed, $q(x|x_a)$ is tractable, see General Author Rebuttal-Derivation Details.) The rationale behind this simplification is that the adversarial sample usually remains in the vicinity of its corresponding clean sample. Moving towards higher log likelihoods of $x$ will probably lead $x_a$ toward the mode of the $x$ with the same ground-truth class label. As discussed in Section 3.2.1, this simplification has its limitations. Therefore, we propose the SR loss to address this issue.
>
> > The method has not been fully tested against ... multiple denoising steps.
>
> (A11) Our proposed ScoreOpt-N method includes gradient-descent step and one-shot-denoising step. Approximating the full gradient using the one-shot denoising surrogate process corresponds to regarding the gradient-descent step as an identity mapping. This approximation method is also employed in [1] and has been empirically shown to be as effective as computing the exact gradient. Since our method doesn't involve multi-step denoising, approximating through multiple denoising steps does not give a more accurate gradient estimate.  In our experiments, we have observed that the success attack rate decreases when the attack differentiates differentiates through multiple denoising steps.
>
> [1] Chen, Huanran, et al. "Robust Classification via a Single Diffusion Model." arXiv preprint arXiv:2305.15241 (2023).

---

> > ### Comment · Reviewer_epqL · 2023-08-14
> >
> > Thank you for addressing most of the comments - this was helpful!
> >
> > I still have few concerns:
> >
> > - (A2): the general comment addresses the issue where $q(x_a | x)$ in Eq. (1) has been updated to $q(x)$. It doesn't however provide a clear definition/separation of $q(x)$ and $q(x_a | x)$, and where $q(x_a)$ needs to be used instead. A clearer definition/separation would be very helpful.
> >
> > - (A3): Do you have some experiments to validate your hypothesis, i.e., where you consider both approaches (taking gradient w.r.t. $x$ and w.r.t. $x_t$) and plot the distribution, and showing that the gradients are indeed more stable?
> >
> > - (A6): I see. The term $\Sigma_z$ was the confusing part. I thought it was a summation over $z$. Can you please update and state that this term is the covariance matrix?
> >
> > - (A10): The adversarial sample is in the vicinity of the clean sample in the pixel space. However, once it is passed to a model, the clean and adversarial outputs are far from each other. I am still not very convinced about the choice. Can you please provide more intuition?
> >
> > Thanks in advance!

---

> > > ### Author Response · Authors · 2023-08-19
> > > **Further Responses to Reviewer epqL**
> > >
> > > Thanks for your valuable feedback. We are glad that we have addressed some of your concerns. In the following paragraphs, we will address other concerns one by one.
> > >
> > > > The general comment addresses the issue where $q(x_a|x)$ in Eq. (1) has been updated to $q(x)$. It doesn't however provide a clear definition/separation of $q(x)$ and $q(x_a|x)$, and where $q(x_a)$ needs to be used instead. A clearer definition/separation would be very helpful.
> > >
> > > (A1) We feel sorry for the confusion. First, let us make some clarification. We use $q(x)$ to approximate the true posterior distribution $p(x|x_a)$, not $q(x_a|x)$. The derivation is based on variational inference. Our motivation is to find a way to determine the true posterior distribution $p(x|x_a)$. However, it is difficult to infer $p(x|x_a)$ directly. The idea behind variational inference to solve this problem is to use a simpler, so tractable distribution $q(x)$ to approximate $p(x|x_a)$. That's why $q(x)$ is used in Eq. (1). And when the evidence upper bound in Eq. (1) is minimized, the KL-divergence between the two distributions, $q(x)$ and $p(x|x_a)$, is also minimized.
> > >
> > > > Do you have some experiments to validate your hypothesis, i.e., where you consider both approaches (taking gradient w.r.t. $x$ and w.r.t. $x_t$) and plot the distribution, and showing that the gradients are indeed more stable?
> > >
> > > (A2) Thank you for the question. It is true that more explorations are favored to verify our hypothesis. However, since the discussion period is short, we only conduct a relatively small experiment to explore the stability of the optimization process. In order to check the stability, we compute the standard deviation of the l2-norm of the gradients in the optimization process. The standard deviation of the l2-norm of the gradients of ScoreOpt-O is 2.07e-4. The standard deviation of ScoreOpt-N is 6.51e-5, smaller than ScoreOpt-O, which shows the stability of the scale of gradients. However, due to the limited time of the discussion period, our exploration may not be very sufficient. We are willing to explore more on this aspect in the future and put the results in the revision.
> > >
> > > > The term $\Sigma_z$ was the confusing part. I thought it was a summation over $z$. Can you please update and state that this term is the covariance matrix?
> > >
> > > (A3) We feel sorry for the confusion. The term $\Sigma_z$ in $\mu_z=z+\Sigma_z \nabla_z \log p(z)$  does represent the covariance matrix of $z$. We will refine it in the revision.
> > >
> > > > The adversarial sample is in the vicinity of the clean sample in the pixel space. However, once it is passed to a model, the clean and adversarial outputs are far from each other. I am still not very convinced about the choice. Can you please provide more intuition?
> > >
> > > (A4) We appreciate your keen intuition. We agree that the classifier outputs of the clean and the adversarial examples are distinct from each other. However, the purpose of our ScoreOpt is to recover the clean sample from the adversarial sample in the pixel space before feeding it into the downstream classifier. Ideally, the final purified sample should remain in the vicinity of the clean sample. And our method is independent of the downstream classifier.
> > >
> > > We hope our answers have resolved your concerns. And if there are any other new questions, please let us know and we are glad to further address them.

---

### Author Rebuttal · Authors · 2023-08-09

We sincerely appreciate the thoughtful comments from all the reviewers. In response to some general concerns raised by the reviewers, we provide a unified reply here. And we address specific questions and feedbacks individually.

**Derivation Details.**

Some of the reviewers mentioned that the notation and derivation of our statements may be a little confusing. To address this concern, we provide more detailed derivations of the statements in Section 3.2 to ensure clarity and better understanding.

We introduce a variational posterior $q(\mathbf{x})$ to approximate the true posterior distribution $p(\mathbf{x}| \mathbf{x}\_a)$ in the original optimization objective. The variational upper bound writes:
$$
-\log p(\mathbf{x}\_a) \leq  \mathbb{E}\_{q(\mathbf{x})}\left[-\log p\left(\mathbf{x}\_a|\mathbf{x}\right)\right] + \operatorname{KL}\left(q(\mathbf{x}) \|\| p\_{\boldsymbol{\theta}} \left(\mathbf{x}\right)\right).
$$
As shown in [1,2], we can obtain an upper bound on the second KL divergence term between the variational posterior and the the prior distribution defined by pre-trained score network.
$$
\operatorname{KL}\left(q(\mathbf{x}) \|\| p\_{\boldsymbol{\theta}} \left(\mathbf{x}\right)\right)
\leq \mathbb{E}\_{q(\mathbf{x})}\mathbb{E}\_{t \sim \mathcal{U}(0,1), \epsilon \sim \mathcal{N}(\mathbf{0}, \mathbf{I})}\left[w(t)\left\|\|s\_{\boldsymbol{\theta}}\left(\mathbf{x}\_t; t\right)-\nabla\_{\mathbf{x}\_t} \log q(\mathbf{x}\_t ) \right\|\|\_2^2\right],
$$
where $w(t)=g(t)^2/2$ and $\mathbf{x}\_t = \mathbf{x} + \sigma\_t \epsilon$.
The simplest approximation to the posterior over $\mathbf{x}$ is point estimate, i.e., the introduced  variational posterior $q(\mathbf{x})$ satisfies the Dirac delta distribution $q(\mathbf{x}) = \delta(\mathbf{x}-\mathbf{x_\mu})$. Thus, the obove upper bound can be rewritten as:
$$
\mathbb{E}\_{t \sim \mathcal{U}(0,1), \epsilon \sim \mathcal{N}(\mathbf{0}, \mathbf{I})}\left[w(t)\left\|\|s\_{\boldsymbol{\theta}}\left(\mathbf{x}\_t; t\right)-\nabla\_{\mathbf{x}\_t} \log q(\mathbf{x}\_t ) \right\|\|\_2^2\right],
$$
where $\mathbf{x}\_t = \mathbf{x\_\mu} + \sigma\_t \epsilon$. We simply use notation $\mathbf{x}$ instead of $\mathbf{x_\mu}$ for convenience in our main paper.
According to Tweedie's formula: $\mu_z=z+\Sigma_z \nabla_z \log p(z)$, we can obtain $\mathbf{x} = \mathbf{x}\_t + \sigma\_t^2 \nabla\_{\mathbf{x}\_t} \log q(\mathbf{x}\_t).$
Defining $D\_{\boldsymbol{\theta}}\left(\mathbf{x}\_t; t\right) \coloneqq \mathbf{x}\_t + \sigma\_t^2 s\_{\boldsymbol{\theta}}\left(\mathbf{x}\_t; t\right)$, the KL term of our opimization objective converts to:
$$
\mathcal{L}(\mathbf{x},\boldsymbol{\theta})=\mathbb{E}\_{t \sim \mathcal{U}(0,1), \epsilon \sim \mathcal{N}(\mathbf{0}, \mathbf{I})}\left[\tilde{w}(t)\left\|\|D\_{\boldsymbol{\theta}}\left(\mathbf{x} + \sigma\_t \epsilon; t\right)-\mathbf{x} \right\|\|\_2^2\right],
$$
where $\tilde{w}(t) = w(t) / \sigma_t^2$.

**Evalutaions.**

There are indeed numerous papers in the adversarial literature, and they are often evaluated using different criteria. In our paper, we have taken great care to ensure a fair comparison. As a result, we compare ScoreOpt with baselines for three types of attacks: transfer-based attacks, strong adaptive attacks BPDA+EOT, and PGD+EOT. The intensity of these attacks increases gradually from weak to strong. It has been demonstrated in [3] that the PGD+EOT attack is stronger than AutoAttack for robust evaluation of diffusion-based methods. Our experimental setups precisely align with those presented in [4], [5], and [3], respectively. Here, we provide a summary of the comparisons between DiffPure and our methods against different attacks.

 | Methods | Transfer | BPDA+EOT | PGD+EOT |
 | ---------- | ---------- | ---------- | ---------- |
 | DiffPure | 90.08| 81.40 | 51.25 |
 | ScoreOpt | 92.30| 90.02 | 65.04 |

**Inference Time.**

As suggested by the reviewers, we provide the final inference times required by our proposed method and diffusion-based baselines to achieve the results presented in Tables 1, 2, and 4. The evaluation of inference speed was conducted on a single NVIDIA TITAN RTX GPU. We present the corresponding inference time results as follows. The reported values represent the final inference times (in seconds) for one successfully defended image.

 | Methods | Transfer $\ell_{\infty}$ |   Transfer $\ell_2$ | BPDA+EOT |
 | ------------ | ------------ | ------------ | ------------ |
 | DiffPure(EDM) | 2.656 | 2.735 | 60.942 |
 | ScoreOpt | 1.934 | 1.981 | 24.966 |

[1] Song, Yang, et al. "Maximum likelihood training of score-based diffusion models." Advances in Neural Information Processing Systems 34 (2021): 1415-1428.

[2] Vahdat, Arash, Karsten Kreis, and Jan Kautz. "Score-based generative modeling in latent space." Advances in Neural Information Processing Systems 34 (2021): 11287-11302.

[3] Lee, Minjong, and Dongwoo Kim. "Robust evaluation of diffusion-based adversarial purification."
arXiv preprint arXiv:2303.09051 (2023).

[4] Yoon, Jongmin, Sung Ju Hwang, and Juho Lee. "Adversarial purification with score-based
generative models." International Conference on Machine Learning. PMLR, 2021.

[5] Nie, Weili, et al. "Diffusion Models for Adversarial Purification." International Conference on
Machine Learning. PMLR, 2022.

---

### Decision · Program_Chairs · 2023-09-21

**Decision:**

Accept (poster)

**Comment:**

This paper received mixed reviews. Some reviewers liked the solid motivation and broad evaluation, while others pointed out issues with the presentation of results and depth of the evaluation. The authors engaged extensively during the review period and addressed most reviewer concerns. I have decided to accept the paper, but the authors *must* incorporate all of the reviewer feedback, including (but not limited to):

- As pointed out by reviewer wKkn, the "top-line" result should be the *minimum* accuracy for a given threat model, not the accuracy for a fixed (weak) attack. The authors addressed this point by showing that their method also performs well on white-box adaptive attacks---*this should be the main result*. I would recommend that the authors move the accuracies against (weak) black-box attacks to the appendix (or later in the paper) and replace them with some of the evaluation done during the rebuttal period.

- The authors should include a comprehensive evaluation of EOT and BPDA in this setting, varying hyperparameters of both attacks and also showing the effectiveness of their attack (it was mentioned in a reply to one of the reviewers, but should be discussed in the main paper).

- The notation and pseudocode should be cleaned up and made consistent, and the derivations from the rebuttal should be included in at least an Appendix of the paper. The writing should generally be proofread and improved to address the concerns of the reviewers below.

These are just a few of the main changes that should be made to the paper---the authors should carefully go through the reviews and responses below and ensure that they make all the appropriate changes.